# Heterogeneous recruitment abilities to RNA polymerases generate nonlinear scaling of gene expression with cell volume

Qirun Wang[1] & Jie Lin [1,2✉]

While most genes' expression levels are proportional to cell volumes, some genes exhibit nonlinear scaling between their expression levels and cell volume. Therefore, their mRNA and protein concentrations change as the cell volume increases, which often have crucial biological functions such as cell-cycle regulation. However, the biophysical mechanism underlying the nonlinear scaling between gene expression and cell volume is still unclear. In this work, we show that the nonlinear scaling is a direct consequence of the heterogeneous recruitment abilities of promoters to RNA polymerases based on a gene expression model at the whole-cell level. Those genes with weaker (stronger) recruitment abilities than the average ability spontaneously exhibit superlinear (sublinear) scaling with cell volume. Analysis of the promoter sequences and the nonlinear scaling of *Saccharomyces cerevisiae*'s mRNA levels shows that motifs associated with transcription regulation are indeed enriched in genes exhibiting nonlinear scaling, in concert with our model.

[1] Center for Quantitative Biology, Academy for Advanced Interdisciplinary Studies, Peking University, Beijing 100871, China. [2] Peking-Tsinghua Center for Life Sciences, Academy for Advanced Interdisciplinary Studies, Peking University, Beijing 100871, China. ✉email: linjie@pku.edu.cn

Homeostasis of gene expression level in a dynamically growing cell volume is widely observed across the domains of biology, namely, the concentrations of mRNAs and proteins of most genes are approximately constant[1–10]. Since the cell volume usually grows exponentially[9,11,12], this implies exponential growth of mRNA and protein copy numbers as well, and a linear scaling between the gene expression level and cell volume exists. Theoretical models have shown that the linear scaling between gene expression level and exponentially growing cell volume is a consequence of the limiting nature of gene expression machinery: RNA polymerases (RNAPs) and ribosomes[13], in agreement with experimental observations[10,14–16]. However, along with the constant concentrations of mRNAs and proteins of most genes, there is a subset of genes exhibiting nonconstant concentrations as the cell volume increases. These genes are often crucial for cell-cycle and size regulation. They allow cells to sense their sizes based on the concentrations of proteins that have different scaling behaviors with cell volume, such as Whi5 and Cln3 in *Saccharomyces cerevisiae*[17–21]. Other examples include DNA-binding proteins such as histone proteins, whose expression levels scale with the total DNA amount instead of cell volume[22]. Recent experiments show that proteins with changing concentration are often associated with cell senescence[23]. A fundamental question then arises: if a linear scaling between gene expression level and cell volume are by default for most genes, how can cells achieve nonlinear scaling for a subset of genes with cell volume in the meantime? In this paper, we show that the superlinear and sublinear scaling of gene expression level is a direct consequence of the heterogeneous recruitment abilities of promoters to RNA polymerases. Given a unimodal distribution of recruitment abilities, those genes with their promoters' recruitment abilities below (above) the average spontaneously exhibit superlinear (sublinear) scaling with cell volume, while genes with recruitment abilities near the average exhibit approximately linear scaling.

In the following, we first introduce a gene expression model at the whole-cell level in which the promoters of all genes have the same recruitment abilities to RNAPs, which is analytically solvable. Then we consider a scenario in which all genes except a small subset of genes have the same recruitment abilities and show that the expression levels of those special genes can exhibit superlinear or sublinear scaling with cell volume depending on their relative magnitudes of recruitment abilities compared with the majority of genes. Then we extend the simplified model to allow a continuous distribution of recruitment abilities and show that our simplified model can quantitatively capture this more realistic scenario. Genes with recruitment abilities below (above) the average value naturally exhibit superlinear (sublinear) scaling with cell volume. Finally, to verify our theoretical predictions, we analyze the nonlinear scaling of mRNA numbers *vs.* cell volume of *S cerevisiae* using the data from Ref. [21]. Our model predicts a positive correlation between the mRNA production rates and nonlinear scaling degrees among genes, and experimental data confirm this. We further analyze the promoter sequences of all genes with measured nonlinear scaling degrees and find that special motifs for transcription factor binding are indeed enriched in the promoters of genes exhibiting nonlinear scaling, in concert with our theoretical predictions. Our results imply that the nonlinear scaling of gene expression level can be under evolutionary selection through the promoter sequences.

## Results

### Model of gene expression at the whole-cell level.
We consider a coarse-grained model of gene expression. The mRNA production rate $k_{n,i}$ of one particular gene labeled by index $i$ is proportional to its gene copy number ($g_i$) and the probability for its promoters to be bound by RNAPs ($P_{b,i}$),

$$k_{n,i} = \Gamma_{n,i} g_i P_{b,i}, \tag{1}$$

where

$$P_{b,i} = \frac{c_{n,free}}{c_{n,free} + K_{n,i}} \tag{2}$$

Here $\Gamma_{n,i}$ is the initiation rate of transcription of gene $i$: the rate that a promoter-bound RNAP starts transcribing the gene and producing mRNA. We assume the probability for one promoter to be bound by an RNAP follows the Michaelis-Menten equation where $c_{n,free}$ is the concentration of free RNAPs in the nucleus[24] (see the schematic in Fig. 1). For simplicity, we consider all the RNAPs to be in the nucleus and neglect the small fractions of RNAP intermediates that may exist in the cytoplasm.

$K_{n,i}$ is the Michaelis-Menten (MM) constant which is inversely proportional to the binding rate of RNAPs on the promoters (see detailed derivations in Supplementary Discussion A). In the following, we use $1/K_{n,i}$ as a metric of the recruitment abilities of genes to RNAPs: a larger $K_{n,i}$ represents a smaller recruitment ability. We assume the mRNA lifetime as $\tau_{m,i}$ so that given the mRNA production rate, the mRNA copy number $m_i$ changes according to Eq. (6) in Methods. Note that we mainly discuss eukaryotic cells in this work in which the transcription and translation processes are spatially separate, and nonspecific binding of RNAPs to DNA are mostly irrelevant[25,26]. As we also mainly discuss the transcription of mRNA, RNAP here refers to RNAP II. Real transcription processes in eukaryotic cells are complex and may involve transcription factors, mediators, enhancers, and TATA-binding proteins[27–29]. Their effects are coarse-grained into the Michaelis-Menten constant within our model, e.g., a transcription factor that increases the expression level of one gene is equivalent to reducing the Michaelis-Menten constant in the RNAP binding probability $P_{b,i}$.

Because the typical time scale for an RNAP to finish transcribing a gene is around one minute[30], much shorter than the typical doubling times, we assume that the dynamics of RNAPs along each gene is in the steady-state. Therefore, the outgoing flux of RNAPs that finish transcribing a gene ($v_n n_i / L_i$) must be equal to the initiation rate of transcription ($P_{b,i}\Gamma_{n,i}$) where $n_i$ is the number of transcribing RNAPs on one copy of gene $i$. Here $L_i$ is the length of the gene $i$ in the number of codons and $v_n$ is the elongation speed of RNAP, which we approximate as a constant. This leads to the expression of $n_i$ as $n_i = P_{b,i}\Lambda_{n,i}$ where $\Lambda_{n,i} = \Gamma_{n,i}L_i/v_n$ is the maximum possible number of transcribing RNAPs on one copy of gene $i$.

A similar model as transcription also applies to the translational process (Fig. 1 and Methods). In the following analysis, we assume that most proteins are nondegradable, and our results regarding nonlinear scaling are equally valid for degradable proteins. The total protein mass of a cell is $M = \sum_i p_i L_i$, in the number of amino acids where $p_i$ is the protein copy number of gene $i$. The total protein mass is known to be proportional to the cell volume[1,31–33]. Therefore, we assume a constant ratio between the total protein mass and cell volume. We further assume that the nuclear volume is proportional to the total cell volume, supported by experimental observations[34].

### A simplified model in which all genes share the same recruitment ability.
In the following, we consider a simplified scenario in which the promoters of all genes have the same recruitment ability to RNA polymerases so that $K_{n,i} = K_n$ for all $i$. Within this scenario, we find that the mass fractions of proteins in the entire proteome are approximately constant as the cell volume increases

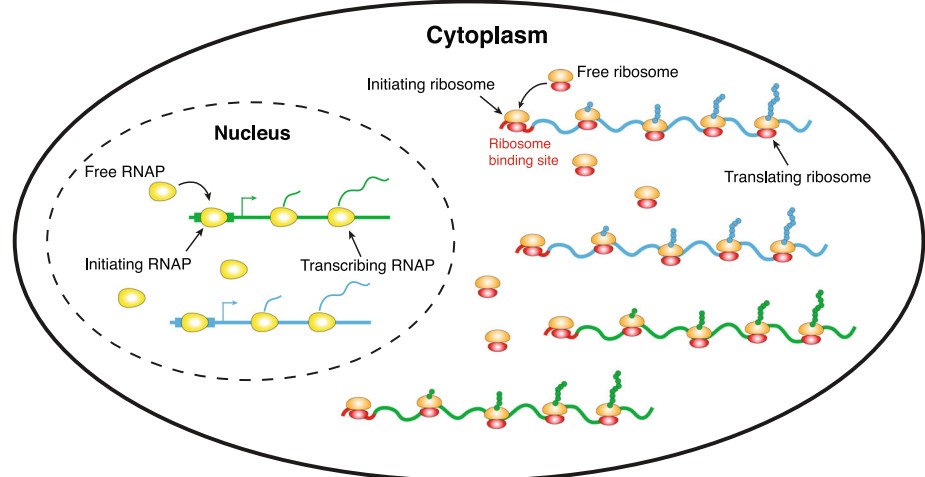

| mRNA production rate of gene i | protein production rate of gene i | cell volume |
|---|---|---|
| $k_{n,i} = \Gamma_{n,i} g_i \dfrac{c_{n,free}}{c_{n,free} + K_{n,i}}$ | $k_{r,i} = \Gamma_{r,i} m_i \dfrac{c_{r,free}}{c_{r,free} + K_{r,i}}$ | $V = V_c + V_n = M/\rho$ <br> $V = aV_n$ |

| | | | | | | |
|---|---|---|---|---|---|---|
| $\Gamma_{n,i}$ | (transcription initation rate) | $\Gamma_{r,i}$ | (translation initation rate) | $V$ (cell volume) | | $M$ (total protein mass) |
| $g_i$ | (gene copy number) | $m_i$ | (mRNA copy number) | $V_c$ (cytoplasmic volume) | | $V_n$ (nuclear volume) |
| $c_{n,free}$ | (free RNAP concentration) | $c_{r,free}$ | (free ribosome concentration) | $\rho$ (protein mass per cell volume) | | |
| $K_{n,i}$ | (MM constant for RNAP binding) | $K_{r,i}$ | (MM constant for ribosome binding) | $a$ (ratio between cell volume and nuclear volume) | | |

**Fig. 1 A coarse-grained model of gene expression at the whole-cell level.** Inside the nucleus, genes compete for free RNAPs to bind to their promoters and for simplicity we only show two genes in this schematic. The binding probability of free RNAPs to the promoters depends on the free RNAP concentration and the recruitment abilities of the promoters. In the cytoplasm, mRNAs compete for free ribosomes to bind to their ribosome binding sites and the binding probability depends on the free ribosome concentrations and the recruitment abilities of the mRNAs to ribosomes. The cell volume $V$, which includes the cytoplasmic volume ($V_c$) and nuclear volume ($V_n$), is proportional to the total protein mass $M$. The ratio between the nuclear volume and cell volume is constant.

given fixed gene copy numbers (Supplementary Discussion B). Therefore, the protein numbers of all genes, including RNAP, are proportional to the total protein mass, and therefore also proportional to the cell volume.

We also assume that almost all RNAPs are bound to a promoter or transcribing, and we will discuss its validity later in this section. Therefore, the total mRNA production rate of all genes is proportional to the total number of RNAPs, which is also proportional to the cell volume. Finally, using the fact that all genes share the same recruitment ability to RNAPs, it follows that the total mRNA production rate should be evenly distributed among all genes. Therefore, each gene's mRNA production rate is also proportional to cell volume, which is the main result of this section.

Here, we also derive the mathematical expression of mRNA production rates. As we show in Methods, the fraction of RNAPs that are bound to a promoter or transcribing is very close to 1 if $n < n_c$ where $n_c = \sum_i g_i(1 + \Lambda_{n,i})$, which is the maximum number of RNAPs the entire genome can hold. In other words, the fraction of free RNAPs, which are neither bound to a promoter nor transcribing, is much smaller than 1 if $n < n_c$. Given this, we obtain the binding probability as $P_b = n/n_c$ (Methods). Using Eq. (1), the mRNA production rate of gene $i$ becomes

$$k_{n,i} = \Gamma_{n,i} \frac{g_i}{n_c} n, \qquad (3)$$

which is proportional to the cell volume.

The above scenario of gene expression, which has been called Phase 1 of gene expression[13], is the typical state of cells in normal conditions[6,14,35] and the main focus of this work. In Phase 1, the cell volume grows exponentially, and the homeostasis of mRNA and protein concentrations of most genes are maintained[13] (see detailed discussions in Supplementary Discussion B). When the total number of RNAPs exceeds the threshold value $n_c$, the linear scaling between mRNA numbers and cell volume breakdowns and cells enter a different phase of gene expression (see detailed discussions in Methods and Supplementary Discussion B).

Finally, since the mRNA lifetimes are typically much shorter than the doubling time[30], mRNA productions are in quick equilibrium. Therefore according to Eq. (6) in Methods, the mRNA copy numbers can be approximated as the products of mRNA production rates and mRNA lifetimes, therefore proportional to the cell volume as well.

**A more realistic model in which genes can have different recruitment abilities.** We now consider a more realistic scenario in which the recruitment abilities to RNAPs of different genes can be different. We start from a simple scenario in which all genes have the same recruitment ability $1/K_n$ except one special gene $i$ has a recruitment ability $1/K_{n,i}$. We note that the only parameter affecting mRNA production rates as the volume changes is the concentration of free RNAPs, $c_{n,free}$, which enters the binding probability $P_{b,i}$ (Eq. (2)). Since the contribution of the particular gene to the global allocation of RNAPs is negligible, the proportionality between the mRNA production rates of most genes and the cell volume is still valid. It then follows that $c_{n,free}$ must change in a way that ensures that the binding probability of

RNAP $P_{b,i}$ is proportional to cell volume for all genes except the particular gene with a different MM constant. Therefore, if the particular gene has a lower MM constant $K_{n,i}$ than the typical $K_n$, this gene is saturated earlier by the rising free RNAP concentration—it thus increases more slowly with increasing volume. A gene with a higher $K_{n,i}$, on the other hand, is not so easily saturated as the typical gene and hence increases superlinearly with volume.

More quantitatively, given the probability for the promoters of most genes to be bound is still $P_b \approx n/n_c$, the free RNAP concentration can be expressed as a function of $n/n_c$. Using the expression of free RNAP concentration, we obtain the mRNA production rate for the particular gene $i$ with the MM constant equal to $K_{n,i}$:

$$k_{n,i} = \Gamma_{n,i} g_i \frac{K_n n}{K_{n,i} n_c - (K_{n,i} - K_n)n}. \qquad (4)$$

As $n$ gets closer to $n_c$ (with $F_n \ll 1$ still satisfied), we find that if $K_{n,i} > K_n$ ($K_{n,i} < K_n$) the mRNA production rate of gene $i$ exhibits a superlinear (sublinear) dependence on the RNAP number. Because the RNAP number is proportional to cell volume, this nonlinear relation is also valid between the mRNA production rates and cell volume, which leads to the nonlinear scaling between the mRNA copy numbers and cell volume. Furthermore, if the corresponding proteins have short lifetimes, their copy numbers will be proportional to the mRNA production rates as well. More importantly, we find that the nonlinear scaling in mRNA copy numbers also propagates to nondegradable proteins, suggesting that the nonlinear scalings in protein copy numbers are insensitive to their lifetimes (Methods).

In the following, we discuss a more realistic scenario that is a continuous distribution of $K_{n,i}$ to reflect the promoter structures' heterogeneity. In this case, we propose that the nonlinear scaling, Eq. (4), is still approximately valid for any gene if $K_n$ is replaced by $\langle K_{n,i} \rangle$, the average value of $K_{n,i}$ among all genes with some appropriate weights. In Supplementary Discussion D, we show that the appropriate weight can be well approximated by the protein mass fractions, as we confirm numerically in the next section. Therefore, those genes with $K_{n,i}$ larger (smaller) than $\langle K_{n,i} \rangle$ exhibit superlinear (sublinear) scaling, while genes with $K_{n,i}$ near the average exhibit approximately linear scaling.

We remark that to ensure the linear scaling of the majority of genes, the RNAP number and the ribosome number should be proportional to the cell volume, which requires that the MM constants of RNAP and ribosome are close to the average value. These are the additional assumptions we make in the case of continuously distributed $K_{n,i}$. For RNAP, it is supported by the constant mRNA concentrations of RNAP related genes observed in the experimental data from Ref. [21] (Supplementary Fig. 10). For ribosome, we found a small deviation of ribosomal mRNA number from linear scaling (Supplementary Fig. 9). However, as we show later, our theoretical predictions on the nonlinear scaling of gene expression level still work satisfyingly well in the presence of a small deviation of the ribosome from linear scaling.

To quantify the nonlinear degree between the expression level and the cell volume of each gene, we introduce a parameter $\beta$ for mRNAs (and also $\alpha$ for nondegradable proteins, see detailed derivations and discussions in Methods), so that its scaling with cell volume becomes

$$\widetilde{m}_i(t) = \widetilde{V}(t) \frac{1 + \beta_i}{1 + \beta_i \widetilde{V}(t)}. \qquad (5)$$

Here the mRNA are measured relative to their values at time $t = 0$, e.g., $\widetilde{m}_i(t) = m_i(t)/m_i(0)$. A similar formula hold for proteins (Eq. (19) in Methods). Positive (negative) $\beta_i$ represent

sublinear (superlinear) scaling behaviors. They are related to the recruitment abilities of their corresponding genes to RNAPs (see the full expressions in Methods) and when $K_{n,i}$ is close to the average Michaelis-Menten constant $\langle K_{n,i} \rangle$, we find that $\beta_i \propto \langle K_{n,i} \rangle - K_{n,i}$.

**Numerical simulations.** We numerically tested our theoretical predictions by simulating a cell with 2000 genes, including RNAP and ribosome, which we coarse-grained as single proteins. Simulation details are summarized in Methods, and parameter values are shown in Supplementary Table 1. To avoid the effects of the cell cycle, which is certainly important but also complicates our analysis, we mainly considered the scenario in which the cell volume grows without cell-cycle progress e.g., cells arrested in the G1 phase, which is a common experimental protocol to study the effects of cell volume on gene expression[21,35]. We also simulated the case of the periodic cell cycle as we discuss later in the Discussion section.

We first simulated the simplified model with homogeneous recruitment abilities to RNAPs. We confirmed the exponential growth of cell volume and the linear scaling between mRNA copy numbers, protein copy numbers, and cell volume (Supplementary Fig. 2). We then simulated the case when all genes share the same recruitment ability $K_n$, but two genes have different abilities that are respectively smaller and larger than $K_n$. Our theoretical predictions for the expression levels of mRNAs, nondegradable proteins (Fig. 2a, b), and degradable proteins (Supplementary Fig. 3), which assume short mRNA lifetimes, match the simulation results reasonably well. We note that the lifetimes of degradable proteins can be comparable to the duration of the cell cycle, e.g., half of the cell-cycle duration[36], in which case deviations of numerical simulations from our theoretical prediction are expected (Supplementary Fig. 3). The nature of nonlinear scaling, whether superlinear or sublinear, is nevertheless independent of the protein's lifetime, as we show in Methods.

Finally, we simulated the more realistic scenario in which $K_{n,i}$ is continuously distributed, which we modeled as a lognormal distribution. Our conclusions are independent of the specific choice of the form of distribution. Examples of volume dependence of several mRNAs and proteins are shown in Supplementary Fig. 4. Nonlinear degrees of mRNAs and proteins are measured based on Eqs. (5, 19).

The resulting distribution of mRNAs is shown in Fig. 3a, and those of proteins are shown in Supplementary Fig. 4. To test our theoretical predictions on the relations between the nonlinear degrees and $K_{n,i}$ (Eqs. (17, 20) in Methods), we also need the expression of the appropriate average $\langle K_{n,i} \rangle$. We found that the appropriate average Michaelis-Menten constant is the one weighted by the initial protein mass fractions, $\langle K_{n,i} \rangle = \sum_i \phi_i K_{n,i}$, where $\phi_i$ is the mass fraction of proteins $i$ in the entire proteome at time $t = 0$. The above expression of $\langle K_{n,i} \rangle$ leads to a good agreement between numerical simulations and theoretical predictions (Fig. 3b). We also used an alternative weight that is the time-averaged protein mass fractions, which works equally well (Supplementary Fig. 5). We set the Michaelis-Menten constants of RNAP and ribosome as the average value to ensure that their concentrations are approximately constant as the cell volume increases.

In Fig. 3a, b, we set the coefficient of variation (standard deviation/mean) of the MM constants as 0.5. To confirm the validity of the model since the recruitment abilities of different promoters can be widely different[27,28,37,38], we also simulated a larger coefficient of variation equal to 1 (Fig. 3c, d). The resulting nonlinear degrees exhibit a broader distribution and appear more similar to experiments as we show in the next section. Furthermore,

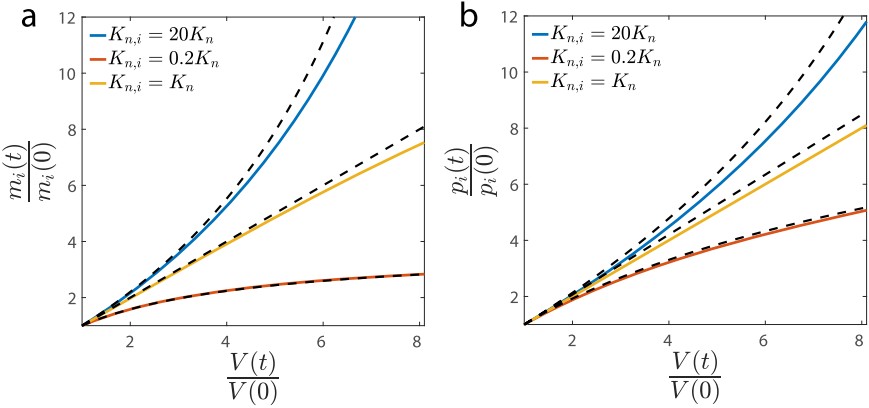

**Fig. 2 We simulate the model with two special genes and their Michaelis-Menten constants are respectively $K_{n,i} = 20K_n$, $K_{n,i} = 0.2K_n$.** All the other genes have $K_{n,i} = K_n$. (**a**) The mRNA levels of the two special genes show superlinear and sublinear scalings with cell volume, in agreement with the theoretical predictions (dashed lines). The mRNA numbers and cell volume are normalized by their initial values. **b** Same analysis as (**a**) for nondegradable proteins. Deviations are expected since the actual mRNA lifetimes are finite.

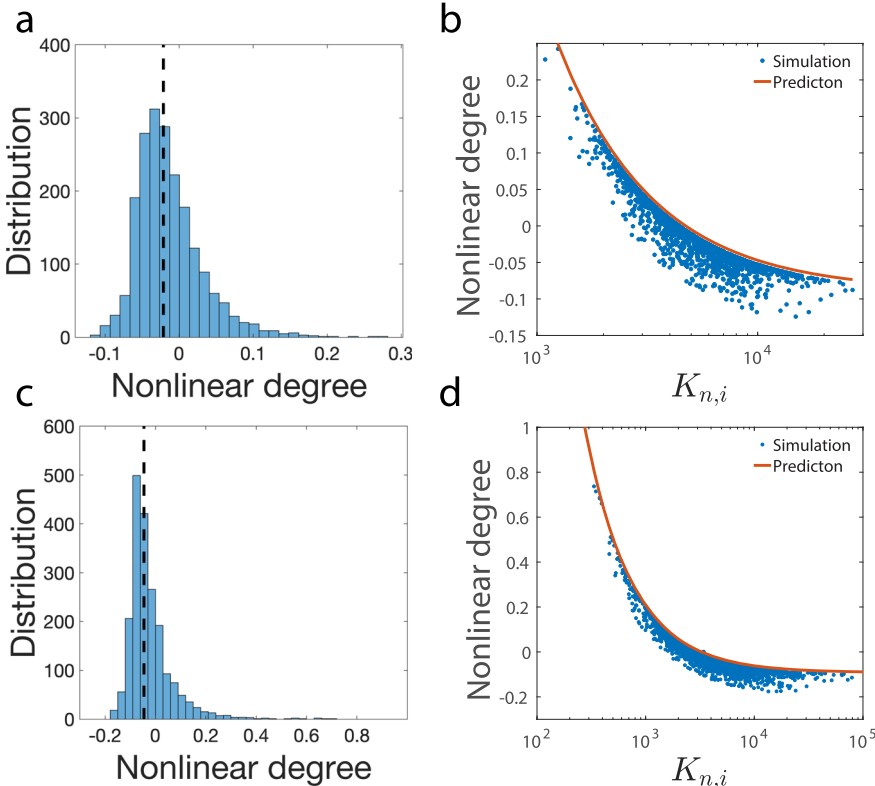

**Fig. 3 Simulation of the scenario in which $K_{n,i}$ is continuously distributed. a** Distribution of the measured nonlinear degrees $\beta$ of mRNA numbers from numerical simulations. The dashed line marks the location of the median value of the nonlinear degrees. **b** We compare the theoretically predicted nonlinear degrees of mRNA numbers and the measured one from numerical simulations. In (**a**) and (**b**), the coefficient of variation of $K_{n,i}$ is 0.5. **c, d** The same analysis as (**a, b**) with the coefficient of variation of $K_{n,i}$ equal to 1.

the theoretical predictions of the nonlinear degrees of mRNA numbers still match reasonably well with the simulations.

We note that the recruitment ability not only determines the nonlinear degree of volume scaling but also affects the mRNA production rate since a higher recruitment ability enhances the binding probability of RNAPs to the promoter. This suggests that there should be a positive correlation between the mRNA production rate and the nonlinear degree. Meanwhile, we note that the recruitment ability also depends on the transcription initiation rate $\Gamma_{n,i}$ (Supplementary Discussion A, Eq. (S2)): a higher initiation rate reduces the recruitment ability (increases the MM

constant). For simplicity, in most of our simulations, we consider a constant $\Gamma_{n,i}$ for genes (except for ribosome and RNAP), and in this case, we indeed found a strong positive correlation between the mRNA production rates and the nonlinear degrees $\beta$ (Supplementary Fig. 4e). However, in a more general model with heterogeneity in $\Gamma_{n,i}$, a higher initiation rate increases the mRNA production rate but also reduces the recruitment ability so that decreases the nonlinear degree. Therefore, heterogeneity in the initiation rates reduces the correlation between the mRNA production rates and the nonlinear degrees. To confirm this prediction, we simulated the case of heterogeneous initiation rates (see numerical details in

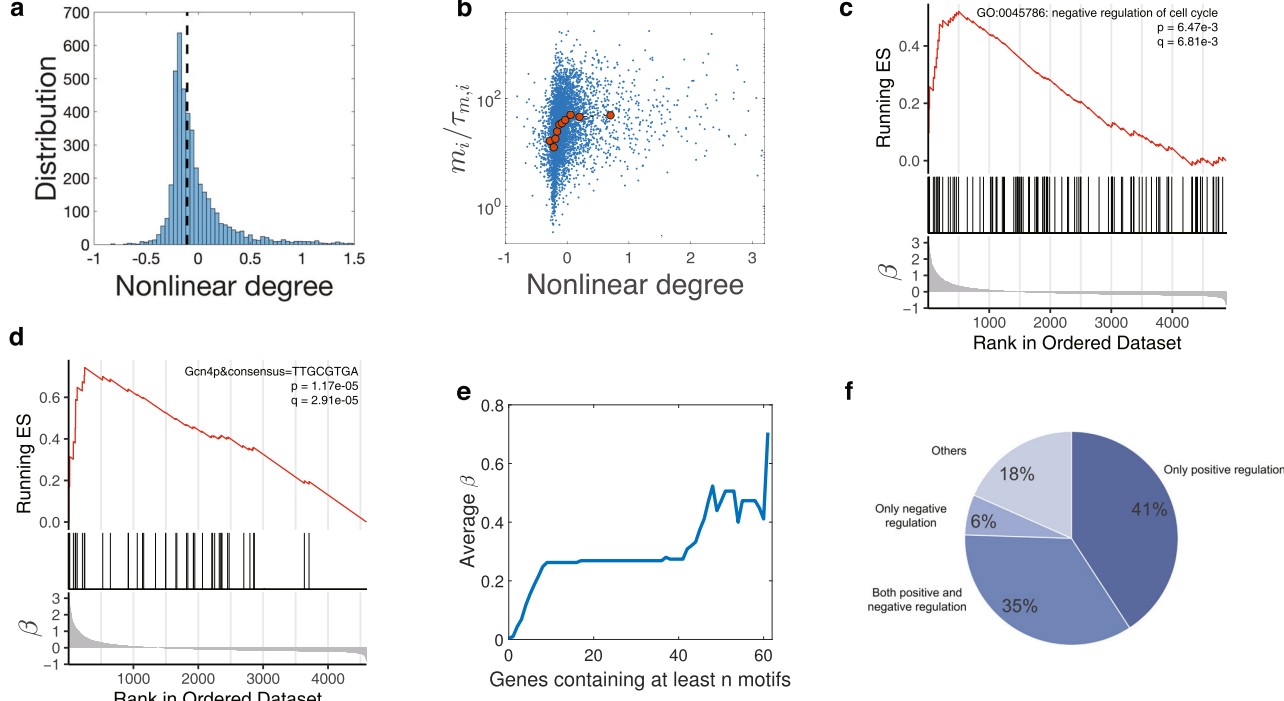

**Fig. 4 Analysis of experimental data. a** Distribution of the nonlinear scaling degrees of mRNAs of *S. cerevisiae* among genes. The dashed line marks the location of the median value of the nonlinear degrees. We consider genes with $-1 < \beta < 3.2$, including 95% of all the measured genes. **b** The Pearson correlation coefficient between the nonlinear degrees of mRNAs and the mRNA production rates is 0.17 (two-sided Pearson correlation test, *p* value < 2.20e-16). The red data points are median values after binning. For the same data, the Spearman correlation coefficient is 0.35 (Spearman correlation test, *p* value < 2.20e-16). **c** Genes annotated as negative regulation of cell cycle are enriched in the sublinear regime. In the bottom panel, genes are ordered by the nonlinear degree $\beta$ from positive (sublinear) to negative (superlinear). In the middle panel, the vertical lines represent the locations of the cell-cycle inhibitors. The upper panel shows the running enrichment score (ES) for the gene set, where the score at the peak is the ES for this gene set. The top-right legend includes the *p* value and the FDR q value of GSEA. **d** An example of a motif that is enriched in the sublinear regime. Here, the vertical lines in the middle panel represent the locations of genes containing the particular motif in the promoter sequences. Note that the motif also appears in the weakly superlinear regime but diminishes in the strongly superlinear regime. **e** We pick out all 77 motifs enriched in the promoters of sublinear genes and calculate the average $\beta$ over genes with at least *n* motifs. **f** The functions of transcription factors associated with the 77 motifs enriched in the sublinear regime. Positive regulation of transcription by RNA polymerase II (GO:0045944) is enriched with **p** value = 2.41e-32. The 76% positive regulation is not likely generated from random sampling (single-sided hypergeometric test, **p** value = 5.29e-4).

Supplementary Discussion E), and our predictions are confirmed numerically (Supplementary Fig. 6). We note that this may be a plausible mechanism of the weak but positive correlation observed in the experimental data, as we discuss in the next section.

Our results suggest that those genes with sublinear scaling and smaller $K_{n,i}$ contribute more in the weighted average of $K_{n,i}$, therefore $\langle K_{n,i} \rangle < \overline{K_{n,i}}$ where $\overline{K_{n,i}}$ is the average over all genes with equal weights. Therefore, genes with $K_{n,i} \approx \overline{K_{n,i}}$ are expected to exhibit superlinear scaling. However, to have an estimation of the nonlinear degree of most genes, the appropriate MM constant to compare with $\langle K_{n,i} \rangle$ is the median value. For the lognormal distribution we used in simulations, we found that the median value of $K_{n,i}$ is close and slightly larger than $\langle K_{n,i} \rangle$. Therefore, the nonlinear degree of the median $K_{n,i}$ is slightly negative compared with the entire distribution (Fig. 3a, c), which is consistent with the experimental observations (Fig. 4a).

**Analysis of experimental data and searching for motifs in the promoter sequences**. We analyzed the genome-wide dataset from Ref. [21] where the volume dependences of mRNA levels are measured for *S. cerevisiae*. We calculated the nonlinear degrees of mRNA scaling with cell volume using Eq. (5) and obtained the resulting distribution (Fig. 4a). The calculated nonlinear degree $\beta$ is highly correlated with the nonlinear scale calculated in Ref. [21]

(Supplementary Fig. 7). The median value of nonlinear degrees is close to zero, suggesting that the majority of genes show approximately linear scaling, similar to our numerical simulations. We found that by choosing appropriate parameters, the numerically simulated distribution of nonlinear degrees matches the experimental measured distribution reasonably well (Supplementary Fig. 8).

We also calculated the correlation between the mRNA production rates and the nonlinear degrees $\beta$. We used the mRNA amount at the smallest cell volume divided by the mRNA lifetime as a proxy for the mRNA production rate according to Eq. (6) (Methods). A weak but significant correlation is indeed observed, as shown in Fig. 4b. As we discussed in the section of numerical simulations, the heterogeneous initiation rates can reduce the correlation between the mRNA production rates and the nonlinear degrees. To further verify our model, we also simulated a modified model in which the nonlinear scaling is independent of the Michaelis-Menten constants (Supplementary Discussion E) and found that the correlation between the mRNA production rates and the nonlinear degrees becomes negative (Supplementary Fig. 6).

We used Gene Set Enrichment Analysis (GSEA)[39,40] to find annotated functional gene sets that are enriched in the superlinear and sublinear scaling regime (Methods, Supplementary Fig. 9). Interestingly, we found that the ribosomal genes and other

translation-related genes, which correspond to the coarse-grained ribosomal genes in our model, are enriched in the superlinear regime with the average $\beta$ over ribosomal genes about $-0.2$. Similar observations were reported in Ref. [21]. However, the superlinear scaling of ribosomal genes was also observed in the control cases in which the nonlinearities of other known nonlinear scaling genes were suppressed. Therefore, it was argued that the superlinear scaling of ribosomal genes may be an artifact due to the drug that blocks cell-cycle progress. Interestingly, recent experiments of mammalian cells found weak sublinear scaling of ribosomal proteins[23]. For RNAP related genes, we found that they indeed show linear scaling with cell volume as we assume in our coarse-grained model, which is crucial for the linear scaling between the mRNA copy numbers and cell volume (Supplementary Fig. 10). To confirm the validity of our conclusions in the presence of weakly superlinear scaling of ribosomes, we numerically simulated our gene expression model with the recruitment ability of ribosomal gene weaker than the average value and found that even with the small deviation of ribosome number from linear scaling, our theoretical predictions still agree well with the numerical simulations (Supplementary Fig. 11).

In Ref. [21], the expression of pre-selected activators for the cell cycle were shown to be superlinear, while pre-selected inhibitors were shown to be sublinear. So we next checked the nonlinear degrees of all cell-cycle regulators using GSEA. We found that inhibitors are indeed enriched in the sublinear regime (Fig. 4c), but activators were not enriched in the superlinear regime. We remark that the inconsistency may be due to the preselection of regulators, but the conclusion of Ref. [21] that the interplay of inhibitors and activators can trigger cell-cycle progress is still valid as long as they have different scaling behaviors.

To further support our theoretical predictions, we investigated the promoter sequences of all genes included in our analysis. We expect that those genes with nonlinear scaling should have some special patterns in their promoter sequences, which render them stronger or weaker recruitment abilities to RNAPs than the others. If this is the case, specific motifs should be enriched in the superlinear or sublinear regime. We detect the transcription factors binding motifs in the promoter sequences and then used GSEA to identify those motifs that are enriched in the nonlinear regime. We found 77 motifs enriched in the sublinear regime (see one example in Fig. 4d and Supplementary Table 2). To further validate our results, we computed the average $\beta$ for genes containing at least $n$ motifs that are enriched in the sublinear regime and found that the average $\beta$ indeed increases as a function of $n$ (Fig. 4e). Consistent with our theoretical predictions, 76% of the 49 corresponding transcription factors exhibit positive regulation and therefore enhance the recruitment abilities to RNAPs of their target genes (Fig. 4f). However, we did not find motifs enriched in the superlinear regime. Considering the cumulative effect of motifs on $\beta$ as shown in Fig. 4e, we propose that antagonistic effect may also exist among motifs, which suggests that motifs reducing the recruitment ability to RNAPs may reside in most genes. But in genes without superlinear scaling, their effects are counteracted by other motifs that enhance the recruitment ability.

## Discussion
For abnormally large cells, the number of RNA polymerases or ribosomes may exceed some threshold values so that the bottlenecks of gene expression become the templates of gene expression: gene copy numbers and mRNA copy numbers[14,35]. However, in typical cellular physiological states, cells are far below the thresholds, and the limiting bottlenecks of gene expression are the copy numbers of RNAPs and ribosomes[10,15,16]. In this case, if the promoters of all genes share the same recruitment ability, the expression levels of all genes should exhibit linear scaling with cell volume both at the mRNA and protein levels[13]. We extended this simple scenario to a more realistic case in which the recruitment abilities among genes are continuously distributed. We derived the dependence of the mRNA production rate on cell volume. We show that genes with recruitment abilities below (above) the average exhibit superlinear (sublinear) scaling with cell volume, a natural consequence of the heterogeneous distribution of recruitment abilities. We further show that the nonlinear scaling between the mRNA production rates and cell volume propagates to the mRNA copy numbers and proteins copy numbers. All of our theoretical predictions were confirmed by numerical simulations.

Nonlinear scaling of protein levels is crucial to cell-cycle regulation[21]. Time-dependent protein concentrations allow cells to determine the timing of various cell-cycle events, e.g., based on the ratio of two proteins with different scaling behaviors. To confirm this scenario, we also simulated the case of the periodic cell cycle and let the cell divide when the ratio of the concentrations of one superlinear protein and one sublinear protein exceeds some threshold value. We found that periodic patterns of mRNA and protein concentration emerge. For superlinear genes, their mRNA and protein concentrations decrease initially at the beginning of the cell cycle due to the halved RNAP number at cell birth, but quickly increases as the RNAP number increases (vice versa for sublinear genes). As the cell gets the periodic steady-state, all mRNAs and proteins double their numbers at cell division compared with cell birth[13,41] (Supplementary Discussion F and Supplementary Fig. 12).

Our model shares some similarities with the model introduced in the Methods section of Ref. [13], but also with key differences. This work focuses on the effects of heterogeneous MM constants and the resulting nonlinear scaling of gene expression levels, including both the mRNA and protein. In contrast, the model in the Methods of Ref. [13] only consider transcription process and homogeneous MM constants. Furthermore, the previous model mainly considers the effects of nonspecifically bound RNAPs, which is believed to be important in bacterial gene expression[24]. The model in this work mainly considers eukaryotic cells, and the experimental data we analyze is from S cerevisiae[21].

The recruitment abilities not only determine the nonlinear degrees of gene expression but also determine the mRNA production rates. Therefore, our model predicts that genes with higher (lower) mRNA production rates are more likely to exhibit sublinear (superlinear) scaling with cell volumes. We also note that heterogeneity in the transcription initiation rates can reduce the positive correlation between the mRNA production rates and the nonlinear scaling degrees, in concert with the small but positive correlation observed in the experimental data of S. cerevisiae[21]. Furthermore, according to our theoretical models, motifs that enhance or reduce the promoters' recruitment abilities should exist in the superlinear or sublinear scaling genes. Indeed, we found a group of motifs enriched in the sublinear regime, and these motifs are associated with transcription factor (TF) binding sites that have positive regulation on the target genes. We note that other mechanisms of nonlinear scaling of gene expression levels are possible, such as time-dependent transcription factor concentrations[21], or time-dependent initiation rates. A time-dependent transcription factor concentration is equivalent to a time-dependent MM constant $K_{n,i}$ within our model. However, we note that our GSEA analysis showed that TF-related terms were not enriched in the nonlinear regime (Supplementary Fig. 9), which means TFs do not change their concentrations in general. Therefore, we argue that a changing TF concentration is

more specific to certain genes instead of a general situation. Also, we remark that our model does not require time-dependent variables to achieve changing concentrations, and the changing concentrations of mRNAs and proteins are the result of the competition between the genes to the limiting resource of RNAPs. We note that in Ref. [21], the numbers of binding sites for specific cell-cycle transcription factors in the target genes were found to be positively correlated with the superlinear degrees of their mRNA levels, which may be due to other mechanisms not related to our model.

Our results can have far-reaching implications: the nonlinear scaling of gene expression level allows cells to sense their sizes based on the ratio of concentrations of different proteins, enabling cells to decide the timing of multiple cell-cycle events such as cell division. Our results suggest that sensing the concentration differences among a group of proteins as a measure of cell volume can be the most accessible option cells can take to achieve cell size regulation. The promoter sequences can also be under evolutionary selection to achieve desired nonlinear scalings of particular genes and robust cell-cycle regulation. Finally, the gene expression model proposed in this work is by construction at the whole-cell level. Therefore, it can be a valuable platform for mathematical modeling of gene expression, especially for problems in which the competition among genes for the limiting resources of RNAPs and ribosomes are crucial.

## Methods
### A summary of the variables used in the main text.

| Variables | Meaning |
|---|---|
| $V$ | cell volume |
| $M$ | totle protein mass |
| $V_c$ | cytoplasmic volume |
| $V_n$ | nuclear volume |
| $\rho$ | protein mass per cell volume |
| $a$ | ratio between cell volume and nuclear volume |
| $k_{n,i}$ | mRNA production rate of gene $i$ |
| $\Gamma_{n,i}$ | transcriptional initiation rate of gene $i$ |
| $g_i$ | gene copy number of gene $i$ |
| $P_{b,i}$ | RNAPs binding probability on the promoter of gene $i$ |
| $c_{n,free}$ | free RNAPs concentration |
| $F_n$ | free RNAPs fraction |
| $\Lambda_{n,i}$ | maximum number of RNAPs one copy of gene $i$ can hold |
| $n_c$ | maximum number of RNAPs the entire genome can hold |
| $L_i$ | length of gene $i$ |
| $v_n$ | RNAP elongation speed |
| $K_{n,i}$ | transcriptional MM constant of gene $i$ |
| $\langle K_{n,i} \rangle$ | weighted average of MM constant |
| $n$ | RNAPs number |
| $\tau_{m,i}$ | lifetime of mRNA $i$ |
| $m_i$ | mRNA number of gene $i$ |
| $k_{r,i}$ | protein production rate of gene $i$ |
| $\Gamma_{r,i}$ | translational initiation rate of gene $i$ |
| $c_r$ | ribosomes concentration |
| $F_r$ | free ribosomes fraction |
| $K_{r,i}$ | translational MM constant of gene $i$ |
| $\tau_{p,i}$ | lifetime of protein $i$ |
| $p_i$ | protein number of gene $i$ |
| $\beta_i$ | nonlinear degree of mRNA $i$ |
| $\alpha_i$ | nonlinear degree of protein $i$ |
| $\phi_n$ | RNAPs mass fraction |
| $\phi_r$ | ribosomes mass fraction |

**Details of the gene expression model.** We explain more details of the gene expression model in this section. Given the mRNA production rate, the mRNA copy number $m_i$ changes as

$$\frac{dm_i}{dt} = k_{n,i} - \frac{m_i}{\tau_{m,i}} \quad (6)$$

where $\tau_{m,i}$ is the mRNA lifetime.

Regarding protein production, the protein production rate of one particular gene is proportional to its corresponding mRNA number ($m_i$), the translation initiation rate ($\Gamma_{r,i}$), and the probability for the ribosome binding site of mRNA to be bound by ribosome:

$$k_{r,i} = \Gamma_{r,i} m_i \frac{c_{r,free}}{c_{r,free} + K_{r,i}}. \quad (7)$$

Here $c_{r,free}$ is the concentration of free ribosomes in the cytoplasm and $K_{r,i}$ is the Michaelis-Menten constant of ribosome binding on the mRNAs (see an alternative formulation of translation model in Ref. [42]).

Because the total number of RNAPs can be separated to free RNAPs, initiating RNAPs, and transcribing RNAPs, we obtained a self-consistent equation to determine the fraction of free RNAPs in all RNAPs:

$$n_c \frac{c_n F_n}{c_n F_n + K_n} = n - nF_n. \quad (8)$$

Here $n$ is the total number of RNAPs, $n_c = \sum_i g_i(1 + \Lambda_{n,i})$, and $F_n$ is the fraction of free RNAPs. $c_n$ is the concentration of total RNAPs in the nucleus and here we consider the simplified model in which all genes have the same MM constant $K_n$ to RNAPs. The left side represents the number of RNAPs bound to promoters or transcribing. The right side represents the difference between the total number of RNAPs and free RNAPs, which should be equal to the left side. Meanwhile, we assume that $c_n \gg K_n$, namely, the total RNAP concentration is much larger than the MM constant of a typical promoter, supported by observations in bacteria[43]. We argue that this assumption is biologically reasonable because if all RNAPs suddenly become free so that $c_{n,free} = c_n$, one would expect that these free RNAPs will have a strong tendency to rebind to the promoters; otherwise, a large fraction of RNAPs will be idle, which is clearly inefficient in normal cellular physiological states. We remark that although the assumption appears reasonable, it remains to be tested in yeast. Using the assumption that $K_n/c_n \ll 1$, we find that the fraction of free RNAPs $F_n$ that solves Eq. (8) must be much smaller than 1 if $n < n_c$ (see the illustration in Supplementary Fig. 13). Therefore, we can take $F_n = 0$ in Eq. (8) and obtain the binding probability as $P_b = n/n_c$.

When the number of RNAPs exceeds the threshold value $n_c$, the linear scaling between the mRNA numbers and cell volume breakdowns for all genes, and the growth mode of cell volume also deviates from exponential growth (Phase 2). If the ribosome number exceeds some threshold value, the cell volume eventually grows linearly, which has been observed in budding yeast[35] (Phase 3, see Ref. [13] and detailed derivations in the Supplementary Discussion B). We note that the main purpose of the assumption $c_n \gg K_n$ is to make the condition of the negligible fraction of free RNAPs more well defined as $n/n_c < 1$. Our conclusions on the relation between the nonlinear scaling and the recruitment abilities do not rely on this assumption. Since we mainly focus on the scenario in which RNAP is limiting with $F_n \ll 1$, the transition details from Phase 1 to Phase 2 is not important to our conclusions. We also discuss the effects of nonspecific binding of RNAPs on the transition between Phase 1 and Phase 2 in Supplementary Discussion C and show that the condition of Phase 1 becomes more stringent in the presence of nonspecific binding.

**Derivation of the nonlinear scaling.** We consider a simple model assuming all genes have the same MM constant $K_n$ except one special gene $i$ has a MM constant $K_{n,i}$. Since the contribution of the particular gene to the global allocation of RNAP is negligible, Eq. (8) is still valid. We focus on Phase 1 so that $F_n \ll 1$ and express $c_n F_n$ as a function of $n/n_c$. The mRNA production rate for the particular gene with the MM constant equal to $K_{n,i}$ therefore becomes Eq. (4).

If $K_{n,i} > K_n$, we find that the mRNA production rate of gene $i$ exhibits a superlinear dependence on the RNAP number, therefore also the cell volume. If $K_{n,i} < K_n$, the mRNA production rate of gene $i$ exhibits a sublinear behavior. The general solution for Eq. (6) becomes

$$m_i(t) = m_i(0)e^{-t/\tau_{m,i}} + \int_0^t e^{-\Delta t/\tau_{m,i}} k_{n,i}(t - \Delta t) d\Delta t. \quad (9)$$

In the limit that the lifetime of mNRA goes to zero, the number of mRNA becomes strictly proportional to the mRNA production rate

$$m_i(t) = \tau_{m,i} k_{n,i}(t). \quad (10)$$

We now consider the dynamics of protein number which is

$$\frac{dp_i}{dt} = \Gamma_{r,i} \frac{c_r F_r}{c_r F_r + K_{r,i}} m_i - \frac{p_i}{\tau_{p,i}}. \quad (11)$$

The general solution for Eq. (11) is

$$p_i(t) = p_i(0)e^{-t/\tau_{p,i}} + \int_0^t e^{-\Delta t/\tau_{p,i}} \Gamma_{r,i} \frac{c_r F_r}{c_r F_r + K_{r,i}} m_i(t - \Delta t) d\Delta t. \quad (12)$$

In the following, we assume $c_r F_r$ to be constant, which is a good approximation in Phase 1 (Supplementary Discussion B) and consider two limiting cases, $\tau_{p,i} \to 0$ and $\tau_{p,i} \to \infty$.

When $\tau_{p,i} \to 0$, the protein number becomes strictly proportional to the protein production rate, which is proportional to the mRNA number,

$$p_i(t) = \Gamma_{r,i} \frac{c_r F_r}{c_r F_r + K_{r,i}} m_i(t) \tau_{p,i}$$
$$= \Gamma_{r,i} \frac{c_r F_r}{c_r F_r + K_{r,i}} \tau_{m,i} \tau_{p,i} k_{n,i}(t). \tag{13}$$

Note that the second identity is valid when $\tau_{m,i} \to 0$ which is a good approximation when the lifetime of mRNA is much shorter than the doubling time.

When $\tau_{p,i} \to \infty$ and $\tau_{m,i} \to 0$ the dynamics of protein number becomes

$$p_i(t) - p_i(0) = \int_0^t \Gamma_{r,i} \frac{c_r F_r}{c_r F_r + K_{r,i}} m_i(t') dt'$$
$$= \Gamma_{r,i} \frac{c_r F_r}{c_r F_r + K_{r,i}} \Gamma_{n,i} g_i \tau_{m,i} \int_0^t \frac{K_n n}{K_{n,i} n_c - (K_{n,i} - K_n) n} dt'. \tag{14}$$

In Phase 1, the number of RNA polymerase increases exponentially $n(t) = n(0) e^{\mu t}$, therefore the integral in Eq. (14) can be analytically calculated

$$p_i(t) - p_i(0)$$
$$= \Gamma_{r,i} \Gamma_{n,i} g_i \tau_{m,i} \frac{c_r F_r}{c_r F_r + K_{r,i}} \frac{K_n}{(K_n - K_{n,i})\mu}$$
$$\times \ln \left( \frac{K_{n,i} n_c - (K_{n,i} - K_n) n(t)}{K_{n,i} n_c - (K_{n,i} - K_n) n(0)} \right). \tag{15}$$

The nonlinear scaling of mRNA copy number also propagates to nondegradable proteins.

To quantify the nonlinear degrees of mRNA copy number, we investigate the volume dependence of mRNA copy number and normalize the volume and mRNA number by their initial values. Using Eqs. (4), (10)), we obtain

$$\widetilde{m}_i(t) = \widetilde{V}(t) \frac{1 - \frac{\Delta K_{n,i} n(0)}{K_{n,i} n_c}}{1 - \frac{\Delta K_{n,i} n(0)}{K_{n,i} n_c} \widetilde{V}(t)}. \tag{16}$$

Here $\widetilde{m}_i(t) = m_i(t)/m_i(0)$, $\widetilde{V}(t) = V(t)/V(0)$, $\Delta K_{n,i} = K_{n,i} - K_n$. We have used the fact that the RNAP concentration is constant therefore $n(t)/n(0) = V(t)/V(0)$. Note that when $K_{n,i}$ is continuously distributed, we should replace $K_n$ by $\langle K_{n,i} \rangle$. Comparing with Eq. (5) in the main text, we find that

$$\beta_i = -\frac{\Delta K_{n,i} n(0)}{K_{n,i} n_c}. \tag{17}$$

When $\Delta K_{n,i} = 0$, $\beta_i = 0$ as expected. For proteins with short lifetimes, the above analysis is equally valid.

We also study the nonlinear degree of nondegradable proteins, and using Eq. (15), we find that

$$\Delta \widetilde{p}_i(t) = C_i \ln \left( 1 - \frac{\Delta K_{n,i}(n(t) - n(0))}{K_{n,i} n_c - \Delta K_{n,i} n(0)} \right)$$
$$= C_i \ln \left( 1 - \frac{\Delta K_{n,i} n(0)}{K_{n,i} n_c - \Delta K_{n,i} n(0)} \Delta \widetilde{V}(t) \right). \tag{18}$$

Here we combine all the constants divided by $p_i(0)$ before the logarithmic term in Eq. (15) to $C_i$ and $\Delta \widetilde{V}(t) = \widetilde{V}(t) - 1$. Therefore, we can write Eq. (18) as

$$\Delta \widetilde{p}_i(t) = C_i \ln \left( 1 + \alpha_i \Delta \widetilde{V}(t) \right), \tag{19}$$

where

$$\alpha_i = -\frac{\Delta K_{n,i} n(0)}{K_{n,i} n_c - \Delta K_{n,i} n(0)}. \tag{20}$$

In the limit $\Delta K_{n,i} \to 0$, we find that

$$\Delta p_i = \Gamma_{r,i} \Gamma_{n,i} g_i \tau_{m,i} \frac{c_r F_r}{c_r F_r + K_{r,i}} \frac{1}{\mu} \frac{n(t) - n(0)}{n_c}, \tag{21}$$

which is consistent with Eq. (14) in the case of $K_{n,i} = K_n$.

**Details of numerical simulations.** All simulations were done in MATLAB (version R2020b and R2021a). We summarize some of the parameters we used in the numerical simulations in Supplementary Table 1. The gene copy numbers are time-independent which we set as 1 for all genes except the ribosome gene which we set as 5. Given an attempted growth rate $\mu_0$, we set the attempted mass fraction of ribosomes in the entire proteome as

$$\phi_r = \mu_0 L_r / v_r. \tag{22}$$

Here, $L_r$ is the length of the ribosome gene in the unit of codons. Note that the actual mass fraction is slightly time-dependent and deviates from the attempted value. To get Eq. (22), we assume that all ribosomes are actively translating and neglect the correction due to free ribosomes and initiating ribosomes. Given the

mass fraction of ribosomes, we assume that the copy number of RNAPs is about 10% of that of ribosomes and set the attempted mass fraction of RNAP as

$$\phi_n = 0.1 \phi_r L_n / L_r. \tag{23}$$

Here $L_n$ is the length of the RNAP gene. In Fig. 2, we set $K_n = 6 \times 10^3 / \mu m^3$, and the two special genes with $K_{n,i} = 20 K_n$ and $K_{n,i} = 0.2 K_n$. We also set the lifetimes of all mRNAs as 10 mins. In Fig. 3, we set $K_{n,i}$ following a lognormal distribution so that it's average $\overline{K_{n,i}} = 6 \times 10^3 / \mu m^3$. We also set the lifetimes of mRNA following a lognormal distribution with a mean equal to 10 min and a coefficient of variation equal to 1. In all simulations, we set the initial attempted total protein mass as $M_b = 10^9$ in the unit of amino acid number and the attempted critical RNAP number as $n_c = \sum_i g_i (1 + \Lambda_{n,i}) = 10^4$.

To find the appropriate value of $\Gamma_{n,i}$ that leads to the attempted $n_c$, we assume $c_n F_n \ll K_{n,i}$ so that

$$\frac{\phi_n}{\phi_r} \approx \frac{\Gamma_{n,n} \frac{g_n}{K_{n,n}} \tau_{m,n} L_n}{\Gamma_{n,r} \frac{g_r}{K_{n,r}} \tau_{m,r} L_r}, \tag{24}$$

from which we can find that $\Gamma_{n,n} = y \Gamma_{n,r}$ where $\Gamma_{n,r}$ and $\Gamma_{n,n}$ are respectively the transcription initiation rates of ribosomes and RNAPs. $y$ can be found using the above equation. We also set $\Gamma_{n,i} = x \Gamma_{n,r}$ for all $i$ except the genes for RNAP ($i = 1$) and ribosome ($i = 2$), so that

$$\phi_r \approx \frac{\Gamma_{n,r} \frac{g_r}{K_{n,r}} \tau_{m,r} L_r}{\Gamma_{n,r} \frac{g_r}{K_{n,r}} \tau_{m,r} L_r + y \Gamma_{n,r} \frac{g_n}{K_{n,n}} \tau_{m,n} L_n + x \Gamma_{n,r} \sum_{i>2} \frac{g_i}{K_{n,i}} \tau_{m,i} L_i}, \tag{25}$$

from which we can find the expression of $x$. Finally, using $\sum_i g_i (1 + \Gamma_{n,i} \frac{L_i}{v_n}) = n_c$, we find that

$$\Gamma_{n,r} = \frac{(n_c - \sum_i g_i) v_n}{g_r L_r + y g_n L_n + x \sum_{i>2} g_i L_i}. \tag{26}$$

from which we find $\Gamma_{n,n}$ and $\Gamma_{n,i}$ for $i > 2$. Note that the approximation $c_n F_n \ll K_{n,i}$ for all $i$ is merely to find the values of $\Gamma_{n,i}$, which do not affect our conclusions.

Before we run the simulations, we use the attempted $\phi_n$, initial cell mass, and $\Gamma_{n,i}$ with $g_i$ and $K_{n,i}$ to calculate the mRNA production rates for all genes using Eqs. (1, 2), from which we use $m_i = k_{n,i} \tau_{m,i}$ as the initial condition. Using the initial $m_i$, we compute the initial protein mass fractions as $\phi_i = m_i L_i / \sum_i m_i L_i$ and also update the mass fractions of RNAP and ribosome. The actual initial mass fractions of RNAP can slightly deviate from the attempted value. To make sure the RNAP copy number is continuous from the beginning of simulations, we also slightly shift the total protein mass from the attempted value so that $\phi_n(\text{actual}) M_b(\text{actual}) = \phi_n(\text{attemped}) M_b(\text{attempted})$.

To remove the transient effects, we take the simulation results at $t = 20$ as the initial values when we compare our results with theoretical predictions. To simulate degradable proteins, we choose 200 of the genes as degradable proteins with lifetimes $\tau_p = 10$ mins and for degradable proteins, we take the simulation results at $t = 100$ as the initial values. The simulation is stopped when the total protein mass is larger than $9M_b$.

**Details of the mRNA production rates.** For the mRNA production rate, we used the mRNA amount (the product of RPKM and cell volume) at the smallest cell volume divided by the mRNA lifetime from Ref. [44] as a proxy according to Eq. (6) in Methods.

**Details of motif searching in the promoters of genes.** Promoters are defined as 500 bp upstream of transcription start sites in the main text. Sequences of promoters for every gene were downloaded from the Yeastract database[45]. Then we used the online tool CentriMo[46] of MEME suite (version 5.3.2)[47] to detect the transcription factors binding motifs annotated in the Yeastract database in these sequences with default parameters.

**Details of GSEA.** We performed enrichment analysis using GSEA using package clusterProfiler (version 3.12.0)[48] in R (version 3.6.1). Genes were ordered by nonlinear degree $\beta$. The cut-off criteria were set as both the $p$ value and the false discovery rate (FDR) q value < 0.05. The number of permutations used in the analysis is 1e5.

In the analysis of functions, the gmt file was generated from KEGG BRITE hierarchy files containing KEGG objects (KO) for budding yeast in Kyoto Encyclopedia of Genes and Genomes (KEGG) database[49–51]. In the analysis of cell-cycle regulators, we generated a gmt file containing only two terms, negative regulation of cell cycle (GO:0045786) and positive regulation of cell cycle (GO:0045787), obtained from Gene Ontology (GO) database with AmiGO (version 2.5.15)[52–54].

In the analysis of motifs, the gmt file was generated based on the CentriMo results. Every motif was considered as a gene set, including multiple genes that contain this motif in their promoters.

**Details of functional enrichment analysis of TFs**. Transcription factors corresponding to the motifs enriched in GSEA were picked out. Functional enrichment analysis for budding yeast was done using Metascape (version 3.5)[55] with default parameters. In 641 genes whose function are regulation of transcription, DNA templated (GO:0006355), 303 genes are annotated as positive regulation of transcription by RNA polymerase II (GO:0045944). In our analysis, all the 49 TFs corresponding to the motifs enriched in GSEA (except one gene ABF2) are annotated as GO:0006355 and 34 of them are annotated as GO:0045944. Based on this situation, single-sided hypergeometric test was performed using R function phyper.

**Reporting Summary**. Further information on research design is available in the Nature Research Reporting Summary linked to this article.

## Data availability

The data that support this study are available from the corresponding author upon reasonable request. All data analyzed in this study are available in publicly accessible repositories. The RNA-seq data (GSE145206) has been published in[21]. Promoter sequences of the yeast genome are available from Yeastract [http://www.yeastract.com/formseqretrieval.php] database[45]. The nonlinear degree data are included in the repository of GSEA codes.

## Code availability

Codes for GSEA are available at https://github.com/QirunWang/R-codes-for-GSEA. Codes for mathematical simulations are available at https://github.com/QirunWang/MATLAB-code-for-simulation.

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

## Acknowledgements

We thank Lucas Carey and Ariel Amir for useful discussions related to this work. We also thank Yuping Chen for kindly sharing experimental data. The research was supported by grants from Peking-Tsinghua Center for Life Sciences.

## Author contributions

J.L. conceived, designed, and carried out the theoretical and numerical part of this work. Q.W. performed the analysis of experimental data and promoter sequences. All the authors contributed to the preparation of the manuscript.

## Competing interests

The authors declare no competing interests
