## [Peer Review File · Nature Communications]

Reviewers' comments:

Reviewer #1 (Remarks to the Author):

In their manuscript "Heterogeneous recruitment abilities to RNA polymerases generate nonlinear scaling of gene expression level with cell volume" Lin and Wang establish a rather general theoretical framework to describe the effect of gene properties on cell-volume-dependent mRNA and protein homeostasis. Analysis of genome wide data reveals a correlation of sublinear scaling of mRNA in budding yeast with mRNA production rates, which is consistent with the conclusion drawn from the theoretical model.

Most protein and mRNA amounts scale with cell volume. However, several recent studies on different organisms revealed that some proteins behave differently, exhibiting either sub- or superlinear scaling. Mathematical modeling (supported by experimental evidence) revealed that this can in principle be explained by a competition of genes for limiting transcriptional machinery, if different genes exhibit different binding affinities for the transcriptional machinery (e.g. Heldt et al., 2018). Focusing on the essence of the mechanism, previous models assumed only a limited number of different gene species. The authors now provide a more generalized modelling framework describing heterogeneous promoter properties. In a first step, the authors then study the limits of identical, and two types of promoters, in which case their model is very similar to that of Heldt et al. The authors then go on to simulate the more general scenario of a log-normal distribution of promoter 'recruitment abilities'.

Mathematical modeling will be crucial to understand scaling of gene expression and protein homeostasis. The establishment of a more general modelling framework allowing for heterogeneous promoter properties to describe scaling of transcript and protein abundance with cell size is definitely a step forward and useful for the field. However, while the initial model is set up in a very general way, the authors (need) to make serious – but in my opinion reasonable assumption – along the way, to obtain their results, which somewhat limits the generality of the conclusions.

Major concerns

1.) In the framework of the model, the gene properties are described by two parameters, the Michaelis Menten constant K , and the initiation rate Γ . While in the initial formulation of the model Γ is specific to each gene, its potential contribution to the scaling behavior is then largely neglected. In particular, for the simulations the authors end up assuming equal Γ for each gene (except polymerase and ribosomes). Importantly however, K depends on Γ (see equation S2), which means that also the scaling behavior depends on Γ . This is critical because it provides an alternative route to achieve sublinear scaling that does not go along with high gene expression: Not only a high binding rate, but also a low initiation rate will result in a low K . However, in this scenario sublinear scaling would correlate with low gene expression. The fact that the analysis of the data by Chen et al. shows a weak correlation between gene expression and sublinear scaling indeed suggests a major role of the binding rate. However, this does not necessarily follow from the general version of the model.

2.) While I appreciate the general formulation of the model, I found it hard to identify the major assumptions made then along the way to reach the conclusion. The assumptions made about Γ described above are one example of this. Another example is the assumption that there are no unbound polymerases made to obtain equation 6. Combined with the assumption that polymerase scales with cell volume, this assumption quite directly leads to the conclusion that mRNA production scales with cell volume as long as $n < n_c$ (without the need for the complex model).

While I appreciate that the authors start with the general model, I think a more upfront communication of the assumptions made in each section would help the reader to understand which assumptions are made to reach the conclusions. Along those lines, the conclusions of the paper are at

times communicated as hard facts, when in fact they depend on assumptions of the model (e.g. in the abstract: ‘... we show that the gene expression scaling is, in fact, ...’)

3.) Another strong assumption of the model is that M and Γ are independent of cell volume. This is not necessarily the case, as for example transcription factor occupancies could change with cell volume (which in fact was proposed by Chen et al. as a potential cause for different size scaling).

4.) If I understood correctly, equation 5 requires that the fraction of free RNAPs in the nucleus and cytoplasm is identical. What is the basis of the assumption? This should be made clearer.

5.) The assumption that $K_n \ll c_n$ is indeed reasonable. However, the justification that it is necessary to obtain the linear scaling seems close to a circular argument. It only ‘follows’ within the framework of the model and the other assumptions.

6.) Although I can see the intuition behind it, the reason why equation 11 holds upon substitution of K_n by the weighted average is not clear to me. Is this based on a formal proof?

Minor suggestions

7.) Final part of the ‘numerical simulations’ section: Why does it not imply that most genes superscale?

8.) Why would the observed superscaling of ribosomes be an artefact of the drug rather than biologically meaningful?

9.) There is a typo in Ref. 4.

Reviewer #2 (Remarks to the Author):

GENERAL REMARKS

This manuscript describes a theoretical model that predicts a linear scaling of mRNA numbers with cell volume for most genes, but a sub-linear (super-linear) scaling for genes with a higher-than-average (lower-than-average) ability to bind RNA polymerase (RNAP). This model is argued to be consistent with experimental data.

The model appears to be very similar to one developed in a previous publication of the first author, Ref. [13], although this relationship is not discussed explicitly. I find the model assumptions questionable, and the agreement with experimental data not convincing (see below).

SPECIFIC COMMENTS

1. It took me a while, but in the end I figured out how to interpret the authors’ model intuitively (please correct me if I’m wrong). I’ll describe it briefly here.

Assumptions:

A1) Virtually all RNAP molecules are always bound to a promoter or transcribing.

A2) Total RNAP concentration is independent of volume.

A3) Almost all genes have the same ability to bind RNAP (i.e., the same $K_{n,i} = K_n$ in a Michaelis-Menten like saturation function P for free RNAP).

Conclusions:

C1) From A1, it immediately follows that the total cellular mRNA production rate k is proportional to

the total number of RNAP molecules, n (not proportional to total RNAP concentration). It then follows from A2 that this rate is proportional to the volume. From A3, it then follows that this rate is evenly distributed among (almost) all genes, i.e., most mRNA species' abundance increases linearly with volume.

C2) The only parameter affecting mRNA production rates k as the volume changes is the concentration of free RNAP, $c_{\{n,free\}}$, which enters the saturation function P . From the proportionality between mRNA production rate k and volume (C1), it then follows that $c_{\{n,free\}}$ must change in a way that ensures that P is proportional to volume; according to A2, this is equivalent to proportionality to n , the total number of RNAP.

C3) If a gene has a lower Michaelis constant $K_{\{n,i\}}$ for free RNAP than the typical K_n , then this gene is saturated earlier by the rising free RNAP concentration $c_{\{n,free\}}$ – it thus increases more slowly with increasing volume. A gene with a higher $K_{\{n,i\}}$, on the other hand, is not so easily saturated as the typical gene, and hence increases super-linearly with volume.

This relatively simple explanation of the authors' mathematical results is currently missing from the manuscript. There is no excuse for that – the mathematical model is fine (and necessary), but it becomes much more useful if the reader is equipped with an intuitive understanding of its behaviour. Unless the authors prefer to address a very specialist audience, I would strongly recommend to even defer the mathematical derivations (almost?) completely to the Methods, and use an intuitive explanation in the main text.

2. Is there a mechanistic reason or experimental evidence for assumption A1 ("Virtually all RNAP molecules are always bound to a promoter or transcribing.")? The thought experiment on page 3 is circular and thus insufficient for this statement.

3. Is there a mechanistic reason for assumption A2("Total RNAP concentration is independent of volume.")? I understand that this is an experimental observation (e.g., on p.3 "Given constant mass fractions of proteins, the total number of RNAPs is proportional to the total protein mass ..."), but the model aims to explain the volume dependence of mRNA and protein concentrations. If the volume dependencies of the two central molecules – RNAP and ribosomes – are assumed as given, then the model's explanatory power appears strongly reduced. This should be discussed.

4. The most problematic assumption appears to be A3 ("Almost all genes have (approximately) the same ability to bind RNAP."). As far as I understand, transcriptional regulation – both on a qualitative and on a quantitative level – occurs to a large extent through binding probabilities of RNAP (e.g., Bintu et al. 2005, DOI: 10.1016/j.gde.2005.02.007). Assuming that all genes have pretty much the same binding probabilities thus seems to fly in the face of what is generally assumed in the field. The authors should devote at least a paragraph to discussing this issue, including a survey of relevant experimental data.

5. Fig. 5A shows that both mode and median of the experimental distribution of the "non-linear degree" of cell volume scaling is clearly below 0, i.e., the typical mRNA shows a super-linear scaling with volume. How does that agree with the authors' model?

6. The authors state that "our model predicts that genes with higher (lower) mRNA production rates [than the typical gene] are more likely to exhibit sublinear (superlinear) scaling with cell volumes". The red points in Fig. 5C show that as a function of the "non-linear degree" (x-axis), the estimated mRNA production rates (y-axis) are minimal at the point where the distribution along the x-axis is most dense (the mode of Fig. 5A) - where the typical gene is located. This appears to contradict the authors interpretation. The authors report an overall correlation between x- and y-axis, which accounts for only $R^2=3\%$ of the observed variation, but do not directly test their prediction. This

omission should be rectified and the result should be discussed appropriately.

7. p.7 "... and then used GSEA to identify those motifs that are enriched in the nonlinear regime with a threshold p value = $5.00e-2$." Did you correct for multiple testing here?

8. The relationship of the authors' model to the very similar model developed in the Methods section of Ref. [13] (Eq. (28) onwards) should be discussed in detail.

MINOR COMMENTS

9. p.4 "...if the corresponding proteins have short lifetimes..." The typical lifetimes in yeast appear to be 1/2 of the cell cycle (Belle et al., PNAS 2006). Is that short enough? Please discuss.

10. p.6 "We note that our results do not contradict with Ref. [21]". How not? Please explain.

11. Methods: I would suggest to restate the definitions of all variables in the Methods section; there are too many of them for the reader to easily memorize these from the main text.

Reviewer #3 (Remarks to the Author):

The paper discusses the scaling relation between protein or mRNA concentrations and cell volume during cell growth. While for many genes an approximately linear scaling is seen, some genes deviate from this. The paper proposes a theoretical explanation for both observations, based on the idea that scaling reflects the ability of the genes to recruit RNA polymerases. The latter is similar for most genes, resulting in the linear scaling, but significant deviations can lead to sub- or super-linear scaling.

I quite like the systematic approach that the authors take, starting from a minimal model, extending it to two subpopulations of genes and then to a continuous distribution of promoter affinities. Finally, a comparison with published data for yeast is done.

Nevertheless I find the overall approach somewhat unsatisfying as it addresses only a part of the cell cycle. In a balanced growth situation, every protein should double over a cell cycle to make sure the two daughter cells have the same content as the mother cell at the beginning of its cell cycle. Likewise for the volume. In terms of concentration, this means that all concentrations at the end of the cell cycle must be the same as at the beginning. Thus any nonlinear scaling of protein/volume or mRNA/volume can only be transient and a superlinear scaling must be followed by a sub-linear one. These aspects have been discussed for the bacterial case (see ref. 13 and also Bierbaum et al. Phys. Biol. 2015) Here only a part of the process is addressed. I am not against doing so, but it is important to keep in mind that this analysis can be design only give a partial picture.

In addition, I have a few more specific comments:

1) Fig 2 is confusing. While I see what you want to do (graphical solution of eq.5), this needs to be explained better, in particular, which parameters are held fixed, what happened to the prefactor of the Michaelis-Menten term?

2) it seems that must be a condition on how n , the number of RNAPs has to grow linearly with the volume for the model to work, which would be a self-consistency requirement. This should in my opinion depend on the degree of binding to DNA. For example for constant initiation rates, a constant free concentration would be required. If all RNAP were free, this would imply a linear scaling with volume (or volume of the nucleus, I am not sure if the latter is assumed to stay constant. If all RNAP

is on DNA, such scaling with volume may not be required

3) likewise ribosomes should also be in the normal group with linear scaling, which seems not to be the case in the experimental data. The authors claim that this does not change their simulation results, but I do not see this. They refer to Fig. S8B, which I think needs to be compared to Fig. S4B and these appear pronouncedly different to me.

4) changes in transcription factor concentrations could provide an alternative explanation for nonlinear scaling. Why can that be completely neglected?

5) p. 5 right column, top: why is the initial mass fraction used as the weight? One could also imagine other quantities to be used, e.g. the average mass fraction

6) fig. 5c: the correlation seen here is very weak. I think it would be good to check with the model whether such weak correlation is sufficient in the model or whether the model leads to stronger correlations.

7) fig. S4B: there seems to be a systematic deviation between the simulation and the prediction, for proteins much more than for the mRNA (Fig. 4D). Why is that?

In my opinion, these points need to be clarified.

List of main changes:

1. As suggested by Reviewer 1 and 2, we have significantly rewritten the part of model descriptions and emphasized the intuitive picture of our model. We have also included a list of variables with definitions in Methods. We believe our revised manuscript is much more accessible to readers.
2. The main changes in the manuscript are highlighted in blue color.
3. We have included new simulations of heterogeneous initiation rates as suggested by Reviewer 1. We found that including heterogeneity in the initiation rates indeed reduces the correlation between the mRNA production rates and nonlinear scaling degrees of gene expression, consistent with experimental observations. The new simulation results are included as Figure S6 in the Supplementary Information, and the details of new simulations are summarized in a new section in the Supplementary Information (section E).
4. We have included new simulations of broader distributions of Michaelis-Menten constants as suggested by Reviewer 2 and found that our theoretical predictions are equally valid. The new simulations are added as Figure 4c,d in the main text of the manuscript.
5. We have included new simulations of periodic cell cycle as suggested Reviewer 3. We found that a robust periodic pattern of gene expression spontaneously emerges in our model. A new figure (Figure S11) is added to the Supplementary Information along with a new section (section F) discussing the simulations of periodic cell cycle.

Reviewer #1 (Remarks to the Author):

In their manuscript “Heterogeneous recruitment abilities to RNA polymerases generate nonlinear scaling of gene expression level with cell volume” Lin and Wang establish a rather general theoretical framework to describe the effect of gene properties on cell-volume-dependent mRNA and protein homeostasis. Analysis of genome wide data reveals a correlation of sublinear scaling of mRNA in budding yeast with mRNA production rates, which is consistent with the conclusion drawn from the theoretical model.

Most protein and mRNA amounts scale with cell volume. However, several recent studies on different organisms revealed that some proteins behave differently, exhibiting either sub- or superlinear scaling. Mathematical modeling (supported by experimental evidence) revealed that this can in principle be explained by a competition of genes for limiting transcriptional machinery, if different genes exhibit different binding affinities for the transcriptional machinery (e.g. Heldt et al., 2018). Focusing on the essence of the mechanism, previous models assumed only a limited number of different gene species. The authors now provide a more generalized modelling framework describing heterogeneous promoter properties. In a first step, the authors then study the limits of identical, and two types of promoters, in which case their model is very similar to that of Heldt et al. The authors then go on to simulate the more general scenario of a log-normal distribution of promoter ‘recruitment abilities’.

Mathematical modeling will be crucial to understand scaling of gene expression and protein homeostasis. The establishment of a more general modelling framework allowing for heterogeneous promoter properties to describe scaling of transcript and protein abundance with cell size is definitely a step forward and useful for the field. However, while the initial model is set up in a very general way, the authors (need) to make serious – but in my opinion reasonable assumption – along the way, to obtain their results, which somewhat limits the generality of the conclusions.

We thank Reviewer 1 for his/her careful reading and appreciating that our work provides a more generalized modelling framework and is definitely a step forward. In the revised manuscript, we have made significant changes and included new simulations of heterogeneous initiation rates as suggested by Reviewer 1.

Major concerns

1.) In the framework of the model, the gene properties are described by two parameters, the Michaelis Menten constant K , and the initiation rate Γ . While in the initial formulation of the model Γ is specific to each gene, its potential contribution to the scaling behavior is then largely neglected. In particular, for the simulations the authors end up assuming equal Γ for each gene (except polymerase and ribosomes). Importantly however, K depends on Γ (see equation S2), which means that also the scaling behavior depends on Γ . This is critical because it provides an alternative route

to achieve sublinear scaling that does not go along with high gene expression: Not only a high binding rate, but also a low initiation rate will result in a low K . However, in this scenario sublinear scaling would correlate with low gene expression. The fact that the analysis of the data by Chen et al. shows a weak correlation between gene expression and sublinear scaling indeed suggests a major role of the binding rate. However, this does not necessarily follow from the general version of the model.

Answer: We appreciate Reviewer 1 for raising this important point and completely agree with Reviewer 1 that the initiation rate Γ affects the Michaelis-Menten (MM) constant K . We remark that the initiation rates can surely be heterogeneous in our model, and in our previous manuscript, we take them constant for each gene (except for RNA polymerase and ribosome) in the numerical simulations just for simplicity. We agree that in a more general model with heterogeneous initiation rates, K and Γ should be correlated.

A larger initiation rate increases the gene expression level, but in the meantime, also increases the MM constant, so that the gene expression becomes more superlinear (a more negative nonlinear degree). Therefore, as suggested by Reviewer 1, heterogeneity in the initiation rates can reduce the correlation between the gene expression levels and the nonlinear degrees. To confirm this prediction, in the revised manuscript, we have added new simulations using the more general expression of the Michaelis-Menten constant $K_{n,i} = \frac{k_{off,i} + \Gamma_{n,i}}{k_{on}}$. Here i labels the index of gene. This expression generates a positive correlation between $K_{n,i}$ and $\Gamma_{n,i}$. Details of numerical simulations are now included in a new section of Supplementary Information (section E).

We found that in this case, a heterogeneity in $\Gamma_{n,i}$ indeed reduces the correlation between the gene expression level (quantified by the mRNA production rate) and the nonlinear degree (see the left and middle panel in the following figure). This also provides a plausible mechanism why the correlation coefficient between the mRNA production rates and nonlinear degrees is weaker in the experimental data (Chen et al., *Molecular Cell*, 2020) compared with our simulations based on the simplified model assuming constant $\Gamma_{n,i}$. We remark that our main conclusion that the nonlinear degree of gene expression scaling depends on $K_{n,i}$ is independent of the correlation between $K_{n,i}$ and $\Gamma_{n,i}$ (see the right panel in the following figure).

Figure 1: (Left) We simulate the more general model in which $K_{n,i} = \frac{k_{off,i} + \Gamma_{n,i}}{k_{on}}$. In this panel, $\Gamma_{n,i}$ is constant (except for ribosome and RNAP). The Pearson correlation coefficient between the mRNA production rate (y axis) and the nonlinear degree is 0.86. (Middle) In this panel, we add heterogeneity to $\Gamma_{n,i}$ so that its coefficient of variation (standard deviation/mean) is 1. The Pearson correlation coefficient is reduced to 0.15, close to the experimental value, 0.17 (Figure 5c in the main text). (Right) We confirm that our main results on the relation between the nonlinear degree and the Michaelis-Menten constant is still valid in the presence of heterogeneity in $\Gamma_{n,i}$.

In the new version of manuscript, we have clarified this important point in the second to last paragraph of the section “Numerical simulations”:

“We note that the recruitment ability not only determines the nonlinear degree of volume scaling but also affects the mRNA production rate since a higher recruitment ability enhances the binding probability of RNAP to the promoter. This suggests that there should be a positive correlation between the mRNA production rate and the nonlinear degree. Meanwhile, we note that the recruitment ability also depends on the transcription initiation rate $\Gamma_{n,i}$ (SI A, Eq. (S2)): a higher initiation rate reduces the recruitment ability (increases the MM constant). For simplicity, in most of our simulations, we consider a constant $\Gamma_{n,i}$ for genes (except for ribosome and RNAP), and in this case, we indeed found a strong positive correlation between the mRNA production rates and the nonlinear degree β (Figure S4c). However, in a more general model with heterogeneity in $\Gamma_{n,i}$, a higher initiation rate increases the mRNA production rate but also reduces the recruitment ability so that decreases the nonlinear degree. Therefore, heterogeneity in the initiation rates reduces the correlation between the mRNA production rates and the nonlinear degrees. To confirm this prediction, we also simulate the case of heterogeneous initiation rates (see numerical details in SI E), and our predictions are confirmed numerically (Figure S6). We note that this may be a plausible mechanism of the weak but positive correlation observed in the experimental data, as we discuss in the next section.”

Along with the new section E in Supplementary Information, we have also included the above figures in the Supplementary Information as Figure S6. We appreciate Reviewer 1 for pointing this, which we believe has significantly improved our manuscript.

2.) While I appreciate the general formulation of the model, I found it hard to identify the major assumptions made then along the way to reach the conclusion. The assumptions made about Γ described above are one example of this. Another example is the assumption that there are no unbound polymerases made to obtain equation 6. Combined with the assumption that polymerase scales with cell volume, this assumption quite directly leads to the conclusion that mRNA production scales with cell volume as long as $n < n_c$ (without the need for the complex model).

While I appreciate that the authors start with the general model, I think a more upfront communication of the assumptions made in each section would help the reader to understand which assumptions are made to reach the conclusions. Along those lines, the conclusions of the paper are at times communicated as hard facts, when in fact they depend on assumptions of the model (e.g. in the abstract: ‘... we show that the gene expression scaling is, in fact, ...’)

Answer: We apologize for this confusion. In the revised manuscript, we have made significant changes to the writing especially in the section “A simplified model in which all genes share the same recruitment ability” and the section “A more realistic model in which genes can have different recruitment abilities”.

In the following, we elaborate on the three assumptions Reviewer 1 raised that need to be clarified:

1. Regarding the assumption of constant initiation rates Γ , we clarify that our main conclusions are independent of the heterogeneity in the initiation rates, but the correlation between the mRNA production rates and the nonlinear degrees can be reduced in a more general model in which the heterogeneity of Γ is considered. See the detailed answer to Question 1.
2. We clarify that the “assumption” that there are no unbound polymerases we use to obtain the linear relation between the mRNA production rate and the RNAP number (now Eq. 3 in the revised version) is a deduction of the assumption that the total RNAP concentration in the nucleus is much larger than the typical Michaelis-Menten constant of RNAP binding ($c_n \gg K_n$). In the revised manuscript, besides a thought experiment, we have also cited relevant references to support our assumption (Bremer, Dennis, Ehrenberg, *Biochimie*, 85, 597-609, 2003), which showed that the typical RNAP concentration is of order $10 \mu M$ while the MM constant is of order $1 \mu M$, therefore $c_n/K_n \sim 10$.

We have now presented our assumptions and predictions in a much more upfront way. In the first paragraph of the section “A simplified model in which

all genes share the same recruitment ability”, we clarify our assumptions and predictions in the beginning:

“In the following, we consider a simplified scenario and assume that (1) the promoters of all genes have the same recruitment ability to RNA polymerases so that $K_{n,i} = K_n$ for all i ; (2) the total RNAP concentration in the nucleus is much larger than K_n , $c_n \gg K_n$ (Bremer et al., *Biochimie*, 2003). As we explain later in this section, the above assumptions lead to two predictions: (1) almost all RNAPs are bound to a promoter or transcribing; (2) the protein mass fractions are constant over time. Therefore the protein numbers of all genes are proportional to the cell volume, including RNAP. From prediction (1), it follows that the total mRNA production rate of all genes is proportional to the total number of RNAPs. From prediction (2), it follows that this rate is also proportional to the cell volume. Finally, combined with assumption (1), it follows that the total mRNA production rate should be evenly distributed among all genes. Therefore, all genes' mRNA numbers increase linearly with volume, which is the main result of this section.”

3. We would like to clarify that the fact that RNA polymerase scales with cell volume is a prediction of our model instead of an assumption in the section “A simplified model in which all genes share the same recruitment ability”. RNA polymerase is treated as every other gene within our model except the fact that RNA polymerase also catalyzes the transcription of all genes including itself. We have now emphasized that the linear scaling between the RNAP number and cell volume is one of the predictions of our model in the first paragraph of the section “A simplified model in which all genes share the same recruitment ability”. The details of the proof are included in Supplementary Information section B.

In the section “A more realistic model in which genes can have different recruitment abilities”, we have also added a paragraph to clarify the assumptions we make in the case of continuously distributed MM constants:

“We remark that to ensure the linear scaling of the majority of genes, the RNAP number and the ribosome number should be proportional to the cell volume, which requires that the MM constants of RNAP and ribosome are close to the average value. These are the additional assumptions we make in the case of continuously distributed $K_{n,i}$. For RNAP, it is supported by the constant mRNA concentrations of RNAP related genes observed in the experimental data from Ref. (Chen et al., *Molecular Cell*, 2020) (Figure S9). For ribosome, we found a small deviation of ribosomal mRNA number from linear scaling (Figure S8). However, as we show later, our theoretical predictions on the nonlinear scaling of gene expression level still work

satisfyingly well in the presence of a small deviation of ribosome from linear scaling.”

We note that our assumption regarding the Michaelis-Menten constant of RNAP is supported by experimental observations that RNAP related genes exhibit constant mRNA concentrations as the cell volume increases (see the following figure, which is shown as Figure S9 in the Supplementary Information).

Figure 2: The RPKM values of RNA polymerase II genes and their average value. The y axis is a good proxy of concentration. There are 52 genes annotated as RNA polymerase II holoenzyme in Gene Ontology database using AmiGO. Error bars represent standard errors. Wilcoxon test results show no significant between-groups differences (V1 vs. V2: $W = 1196$, $p \text{ value} = 3.12e-1$; V2 vs. V3: $W = 1390$, $p \text{ value} = 8.07e-1$; V3 vs. V4: $W = 1357$, $p \text{ value} = 9.77e-1$; V4 vs. V5: $W = 1312$, $p \text{ value} = 7.97e-1$).

3.) Another strong assumption of the model is that M and Γ are independent of cell volume. This is not necessarily the case, as for example transcription factor occupancies could change with cell volume (which in fact was proposed by Chen et al. as a potential cause for different size scaling).

Answer: We apologize for the confusion here. We did not intend to make such a strong assumption as M in our paper represents the total protein mass of the cell, which is proportional to the cell volume and grows over time.

Regarding the initiation rate Γ , we think that the changing transcription factor (TF) occupancies can be modelled as a volume-dependent or time-dependent Michaelis-Menten constant within our model. We completely agree with Reviewer 1 that this could be alternative mechanism to achieve a nonlinear scaling. In this work,

we choose to focus on a simple scenario in which genes' Michaelis-Menten constants to RNAP binding are different but do not change over time.

We would like to mention that in our GSEA (Gene Set Enrichment Analysis) analysis, we found that TF related terms were not enriched in the nonlinear regime (see the figure below for the annotated functional gene sets in KEGG that are enriched in the nonlinear scaling regime, which is shown as Figure S8 in the Supplementary Information), which means TFs do not change their concentrations in general.

Figure 3: Functional gene sets enriched in the nonlinear scaling regime. GeneRatio represents tags in GSEA, which is the fraction of leading-edge genes in those genes that are both in our list of genes and in the corresponding functional gene sets of KEGG. Point size represents the number of leading-edge genes. Colors of the points represent the adjusted p value (FDR). Names of the gene sets are followed by their IDs in KEGG data base (Kanehisa and Goto, *Nucleic Acids Research*, 2000, Kanehisa, *Protein Science*, 2019, Kanehisa et al., *Nucleic Acids Research*, 2020).

Therefore, we argue that a changing TF concentration could be more specific to certain genes instead of a general situation. In the revised manuscript, we have clarified this point in the Discussion section:

“We note that other mechanisms of nonlinear scaling of gene expression levels are possible, such as time-dependent transcription factor concentrations (Chen et al., *Molecular Cell*, 2020), or time-dependent initiation rates. A time-dependent transcription factor concentration is equivalent to a time-dependent MM constant $K_{n,i}$ within our model. However, we note that our GSEA analysis showed that TF related terms were not enriched in the nonlinear regime (Figure S8), which means TFs do not change their concentrations in general. Therefore, we argue that a changing TF concentration is more specific to certain genes instead of a general situation. Also, we remark that our model does not require time-dependent variables to achieve changing

concentrations, and the changing concentrations of mRNAs and proteins are the result of the competition between genes to the limiting resource of RNAPs.”

4.) If I understood correctly, equation 5 requires that the fraction of free RNAPs in the nucleus and cytoplasm is identical. What is the basis of the assumption? This should be made clearer.

Answer: We apologize for this confusion. In the revised manuscript, we have emphasized that all the RNAPs are in the nucleus within our model, and the nuclear volume is proportional to the cell volume. See changes in the first paragraph of the section “Model of gene expression at the whole-cell level”:

“For simplicity, we consider all the RNAPs to be in the nucleus and neglect the small fractions of RNAP intermediates that may exist in the cytoplasm.”

And also the last paragraph of the same section:

“We further assume that the nuclear volume is proportional to the total cell volume, supported by experimental observations (Neumann & Nurse, *The Journal of Cell Biology*, 2007).”

5.) The assumption that $K_n \ll c_n$ is indeed reasonable. However, the justification that it is necessary to obtain the linear scaling seems close to a circular argument. It only ‘follows’ within the framework of the model and the other assumptions.

Answer: See the answer to Question 2.

6.) Although I can see the intuition behind it, the reason why equation 11 holds upon substitution of K_n by the weighted average is not clear to me. Is this based on a formal proof?

Answer: We thank Reviewer 1 for raising this question which let us rethink this approximation. In the revised manuscript, we have now added an argument that the substitution of K_n by the weighted average is a good approximation. We have clarified this point in the third paragraph of the section “A more realistic model in which genes can have different recruitment abilities”:

“In this case, we propose that the nonlinear scaling, Eq. (4), is still approximately valid for any gene if K_n is replaced by $\langle K_{n,i} \rangle$, the average value of $K_{n,i}$ among all genes with some appropriate weights. In SI D, we show that the appropriate weight can be well approximated by the protein mass fractions, as we confirm numerically in the next section.”

The argument is now included in the Supplementary Information section D.

Furthermore, as asked by Reviewer 3, besides the initial protein mass fractions, we also investigate alternative weight such as the protein mass fractions averaged over time. We find that this alternative weight works equally well (see the following figures).

Figure 4: We compare the theoretically predicted nonlinear degrees of mRNA numbers and the measured one from numerical simulations. In both panels, the coefficient of variation of the Michaelis-Menten constants is 1. (Left) The average MM constant is computed as a weighted average over the initial protein mass fractions. (Right) The same simulations as the left panel, but the weight is based on the time-averaged protein mass fractions over the total duration of simulations.

The above figures are now included as Figure S5 in the Supplementary Information.

Minor suggestions

7.) Final part of the ‘numerical simulations’ section: Why does it not imply that most genes superscale?

Answer: We apologize for this confusion, and we have rewritten the paragraph to be more clear. Because sublinear scaling proteins contribute more to the weighted average, $\langle K_{n,i} \rangle < \overline{K_{n,i}}$, where $\overline{K_{n,i}}$ is the average over all genes with equal weights. However, to have an estimation of the nonlinear degree of most genes, the appropriate MM constant to compare with $\langle K_{n,i} \rangle$ is the median value. For lognormal distribution, we find that the median value of $K_{n,i}$ is close and slightly larger than $\langle K_{n,i} \rangle$. Therefore, the nonlinear degree for the median $K_{n,i}$ is slightly negative compared with the entire distribution of nonlinear degrees, which is what we observed in numerical simulations. Below we show two distributions of the nonlinear degrees of mRNA number from our simulations. In the left panel (Figure 4a in the main text of manuscript), the coefficient of variation of $K_{n,i}$ is equal to 0.5, and in the right panel (Figure 4c in the main text of the manuscript), the coefficient of variation is 1.

Figure 5: (Left) Distribution of the measured nonlinear degrees β of mRNA numbers from numerical simulations. The dashed line marks the location of the median value of the nonlinear degrees. Here, the coefficient of variation of $K_{n,i}$ is equal to 0.5. (Right) The same analysis as the left panel with the coefficient of variation equal to 1.

This is consistent with the experimentally measured nonlinear degrees (see the following figure, which is Figure 5a in the main text).

Figure 6: Distribution of the nonlinear scaling degrees of mRNAs of *S. cerevisiae* among genes. The dashed line marks the location of the median value of the nonlinear degrees.

In the revised manuscript, we have rephrased the last paragraph in the section “Numerical simulations”:

“Our results suggest that those genes with sublinear scaling and smaller $K_{n,i}$ contribute more in the weighted average of $K_{n,i}$, therefore $\langle K_{n,i} \rangle < \overline{K_{n,i}}$ where $\overline{K_{n,i}}$ is the average over all genes with equal weights. Therefore, genes with $K_{n,i} \approx \overline{K_{n,i}}$ are expected to exhibit superlinear scaling. However, to have an estimation of the nonlinear degree of most genes, the appropriate MM constant to compare with $\langle K_{n,i} \rangle$ is the median value. For the lognormal distribution we used in simulations, we found that the median value of $K_{n,i}$ is close and slightly larger than $\langle K_{n,i} \rangle$. Therefore, the nonlinear degree of the median $K_{n,i}$ is slightly negative compared with the entire distribution (Figure 4a, c), which is consistent with the experimental observations (Figure 5a).”

8.) Why would the observed superscaling of ribosomes be an artefact of the drug rather than biologically meaningful?

Answer: We would like to clarify that the potential artifact of the super-scaling of ribosomal genes due to the drug (1NMPP1) is the conclusion of the authors of the original experimental paper (Chen et al., *Molecular Cell*, 2020) as they mentioned in their paper: “the positive size-scaling slope of the ribosomal genes may be a surprising but specific response to 1NMPP1, since it occurs even in CDC28 strains without elutriation.” They found that the superlinear scaling of ribosomal genes is still present in the control experiments while the nonlinearities of other known nonlinear scaling genes are suppressed.

In the revised version, we have elaborated on this point in the first paragraph of the section “Analysis of experimental data and searching for motifs in the promoter sequences”:

“Interestingly, we found that the ribosomal genes and other translation-related genes, which correspond to the coarse-grained ribosomal genes in our model, are enriched in the superlinear regime with the average β over ribosomal genes about -0.2 . Similar observations were reported in Ref. (Chen et al., *Molecular Cell*, 2020). However, the superlinear scaling of ribosomal genes was also observed in the control cases in which the nonlinearities of other known nonlinear scaling genes were suppressed. Therefore, it was argued that the superlinear scaling of ribosomal genes may be an artifact due to the drug that blocks cell-cycle progress. For RNAP related genes, we found that they indeed show linear scaling with cell volume as we assume in our coarse-grained model, which is crucial for the linear scaling between the mRNA copy numbers and cell volume (Figure S9). To confirm the validity of our conclusions in the presence of weakly superlinear scaling of ribosomes, we numerically simulated our gene expression model with the recruitment ability of ribosomal gene weaker than the

average value and found that even with the small deviation of ribosome number from linear scaling, our theoretical predictions still agree well with the numerical simulations (Figure S10).”

Finally, we remark that even we take account of the small deviation of ribosome number from linear scaling into our simulations, our theoretical predictions regarding the nonlinear degrees of mRNAs and proteins still work satisfyingly well (see the following figure, which is Figure S10 in the Supplementary Information).

Figure 7: Numerical simulations of the full model in which $K_{n,r}$ is larger than $\langle K_{n,i} \rangle$. The nonlinear degree β of the ribosome gene is about -0.2 . (Left) We compare the theoretically predicted nonlinear degrees of mRNA numbers and the measured values from numerical simulations. (Right) We compare the theoretically predicted nonlinear degrees of protein numbers and the measured values from numerical simulations.

9.) There is a typo in Ref. 4.

Answer: We have fixed this.

Reviewer #2 (Remarks to the Author):

GENERAL REMARKS

This manuscript describes a theoretical model that predicts a linear scaling of mRNA numbers with cell volume for most genes, but a sub-linear (super-linear) scaling for genes with a higher-than-average (lower-than-average) ability to bind RNA polymerase (RNAP). This model is argued to be consistent with experimental data.

The model appears to be very similar to one developed in a previous publication of the first author, Ref. [13], although this relationship is not discussed explicitly. I find the model assumptions questionable, and the agreement with experimental data not convincing (see below).

We thank Reviewer 2 for his/her careful reading and the extremely useful intuitive argument. In the revised manuscript, we have made significant changes to the writing and included the suggested intuitive arguments along with other changes that we believe have significantly improved our paper.

SPECIFIC COMMENTS

1. It took me a while, but in the end I figured out how to interpret the authors' model intuitively (please correct me if I'm wrong). I'll describe it briefly here.

Assumptions:

- A1) Virtually all RNAP molecules are always bound to a promoter or transcribing.
- A2) Total RNAP concentration is independent of volume.
- A3) Almost all genes have the same ability to bind RNAP (i.e., the same $K_{n,i} = K_n$ in a Michaelis-Menten like saturation function P for free RNAP).

Conclusions:

C1) From A1, it immediately follows that the total cellular mRNA production rate k is proportional to the total number of RNAP molecules, n (not proportional to total RNAP concentration). It then follows from A2 that this rate is proportional to the volume. From A3, it then follows that this rate is evenly distributed among (almost) all genes, i.e., most mRNA species' abundance increases linearly with volume.

C2) The only parameter affecting mRNA production rates k as the volume changes is the concentration of free RNAP, $c_{n,free}$, which enters the saturation function P . From the proportionality between mRNA production rate k and volume (C1), it then follows that $c_{n,free}$ must change in a way that ensures that P is proportional to volume; according to A2, this is equivalent to proportionality to n , the total number of RNAP.

C3) If a gene has a lower Michaelis constant $K_{n,i}$ for free RNAP than the typical K_n , then this gene is saturated earlier by the rising free RNAP concentration $c_{n,free}$ – it thus increases more slowly with increasing volume. A gene with a

higher $K_{n,i}$, on the other hand, is not so easily saturated as the typical gene, and hence increases super-linearly with volume.

This relatively simple explanation of the authors' mathematical results is currently missing from the manuscript. There is no excuse for that – the mathematical model is fine (and necessary), but it becomes much more useful if the reader is equipped with an intuitive understanding of its behaviour. Unless the authors prefer to address a very specialist audience, I would strongly recommend to even defer the mathematical derivations (almost?) completely to the Methods, and use an intuitive explanation in the main text.

Answer: We thank Reviewer 2 for this extremely useful comment. We completely agree that adding simple explanations of our results will be very helpful for readers to understand our model better. In the revised version, we have significantly reduced the mathematical derivations in the main text and emphasized the intuitive arguments suggested by Reviewer 2.

In the section “A simplified model in which all genes share the same recruitment ability”, we have significantly changed the structure and presented the simple explanation in the first paragraph:

“In the following, we consider a simplified scenario and assume that (1) the promoters of all genes have the same recruitment ability to RNA polymerases so that $K_{n,i} = K_n$ for all i ; (2) the total RNAP concentration in the nucleus is much larger than K_n , $c_c \gg K_n$ (Bremer et al., *Biochimie*, 2003). As we explain later in this section, the above assumptions lead to two predictions: (1) almost all RNAPs are bound to a promoter or transcribing; (2) the protein mass fractions are constant over time. Therefore the protein numbers of all genes are proportional to the cell volume, including RNAP. From prediction (1), it follows that the total mRNA production rate of all genes is proportional to the total number of RNAPs. From prediction (2), it follows that this rate is also proportional to the cell volume. Finally, combined with assumption (1), it follows that the total mRNA production rate should be evenly distributed among all genes. Therefore, all genes' mRNA numbers increase linearly with volume, which is the main result of this section.”

In the section “A more realistic model in which genes can have different recruitment abilities”, we have also reordered the structure and presented the intuitive arguments in the beginning:

“We now consider a more realistic scenario in which the recruitment abilities to RNAPs of different genes can be different. We start from a simple scenario in which all genes have the same recruitment ability $1/K_n$ except one special gene i has a recruitment ability $1/K_{n,i}$. We note that the only parameter affecting mRNA production rates as the volume changes is the concentration of free RNAPs, $c_{n,free}$,

which enters the binding probability $P_{b,i}$ (Eq. (2)). Since the contribution of the particular gene to the global allocation of RNAPs is negligible, the proportionality between the mRNA production rates of most genes and the cell volume is still valid. It then follows that $c_{n,free}$ must change in a way that ensures that the binding probability of RNAP $P_{b,i}$ is proportional to cell volume for all genes except the particular gene with a different MM constant. Therefore, if the particular gene has a lower MM constant $K_{n,i}$ than the typical K_n , then this gene is saturated earlier by the rising free RNAP concentration -- it thus increases more slowly with increasing volume. A gene with a higher $K_{n,i}$, on the other hand, is not so easily saturated as the typical gene and hence increases superlinearly with volume.”

Finally, we would like to have some additional discussions on Reviewer 2’s assumptions and conclusions to make sure our results are clear.

A1) Virtually all RNAP molecules are always bound to a promoter or transcribing.

We agree with Reviewer 2 that this condition is very important in our model to explain the linear scaling between mRNAs and cell volume. We would like to clarify that this is not the original assumption we make but a direct conclusion of the relation between the total RNA polymerase (RNAP) concentration and the Michaelis-Menten (MM) constant of RNAP binding (see the answer to Comment 2), as we explain in the second paragraph of the section “A simplified model in which all genes share the same recruitment ability”.

A2) Total RNAP concentration is independent of volume.

We mostly agree with Reviewer 2, but we also have some clarifications on this assumption (see the answer to Comment 3).

A3) Almost all genes have the same ability to bind RNAP (i.e., the same $K_{n,i}=K_n$ in a Michaelis-Menten like saturation function P for free RNAP).

Answer: We apologize for this confusion. We would like to clarify that we assume all genes share the same Michaelis-Menten constant only in the section “A simplified model in which all genes share the same recruitment ability”. In the section “A more realistic model in which genes can have different recruitment abilities”, we consider a continuous distribution of $K_{n,i}$, which does not have to be a narrow distribution. We explain this in more detail in the answer to Comment 4.

Conclusions:

C1) From A1, it immediately follows that the total cellular mRNA production rate k is proportional to the total number of RNAP molecules, n (not proportional to total RNAP concentration). It then follows from A2 that this rate is proportional to the

volume. From A3, it then follows that this rate is evenly distributed among (almost) all genes, i.e., most mRNA species' abundance increases linearly with volume.

Answer: We agree with Reviewer 2 on this conclusion.

C2) The only parameter affecting mRNA production rates k as the volume changes is the concentration of free RNAP, $c_{n,\text{free}}$, which enters the saturation function P . From the proportionality between mRNA production rate k and volume (C1), it then follows that $c_{n,\text{free}}$ must change in a way that ensures that P is proportional to volume; according to A2, this is equivalent to proportionality to n , the total number of RNAP.

Answer: We agree with Reviewer 2 on this conclusion.

C3) If a gene has a lower Michaelis constant $K_{n,i}$ for free RNAP than the typical K_n , then this gene is saturated earlier by the rising free RNAP concentration $c_{n,\text{free}}$ – it thus increases more slowly with increasing volume. A gene with a higher $K_{n,i}$, on the other hand, is not so easily saturated as the typical gene, and hence increases super-linearly with volume.

Answer: We agree with Reviewer 2 on this conclusion, which is also what we have in mind. We thank Reviewer 2 for making this point explicitly, which we believe makes our manuscript much more accessible to readers.

2. Is there a mechanistic reason or experimental evidence for assumption A1 (“Virtually all RNAP molecules are always bound to a promoter or transcribing.”)? The thought experiment on page 3 is circular and thus insufficient for this statement.

Answer: In the revised manuscript, besides a thought experiment, we have also cited relevant references to support our assumption (Bremer, Dennis, Ehrenberg, *Biochimie*, 85, 597-609, 2003), which showed that the typical RNAP concentration (c_n) is of order $10 \mu\text{M}$ while the typical MM constant (K_n) is of order $1 \mu\text{M}$, therefore $c_n/K_n \sim 10$. This leads to the conclusion that most RNAPs are bound to a promoter or transcribing.

3. Is there a mechanistic reason for assumption A2 (“Total RNAP concentration is independent of volume.”)? I understand that this is an experimental observation (e.g., on p.3 “Given constant mass fractions of proteins, the total number of RNAPs is proportional to the total protein mass ... “), but the model aims to explain the volume dependence of mRNA and protein concentrations. If the volume dependencies of the two central molecules – RNAP and ribosomes – are assumed as given, then the model's explanatory power appears strongly reduced. This should be discussed.

Answer: We would like to clarify that the linear scaling between the RNAP, ribosome number, and cell volume is a prediction of our model given the assumption that all the genes share the same Michaelis-Menten constant of RNAP binding, as we discuss in

the section “A simplified model in which all genes share the same recruitment ability”. The protein mass fractions are always constant within this scenario and largely determined by the gene copy numbers, as we discuss in the Supplementary Information section B. We have now emphasized that the linear scaling between the RNAP number and cell volume is one of the predictions of our model in the first paragraph of the section “A simplified model in which all genes share the same recruitment ability”:

“In the following, we consider a simplified scenario and assume that (1) the promoters of all genes have the same recruitment ability to RNA polymerases so that $K_{n,i} = K_n$ for all i ; (2) the total RNAP concentration in the nucleus is much larger than K_n , $c_n \gg K_n$ (Bremer et al., *Biochimie*, 2003). As we explain later in this section, the above assumptions lead to two predictions: (1) almost all RNAPs are bound to a promoter or transcribing; (2) the protein mass fractions are constant over time. Therefore the protein numbers of all genes are proportional to the cell volume, including RNAP.”

In the section “A more realistic model in which genes can have different recruitment abilities”, we agree that the constant total RNAP concentration and ribosome concentration are important assumptions and we have now added a paragraph to clarify the assumptions we make in the case of continuously distributed MM constants:

“We remark that to ensure the linear scaling of the majority of genes, the RNAP number and the ribosome number should be proportional to the cell volume, which requires that the MM constants of RNAP and ribosome are close to the average value. These are the additional assumptions we make in the case of continuously distributed $K_{n,i}$. For RNAP, it is supported by the constant mRNA concentrations of RNAP related genes observed in the experimental data from Ref. (Chen et al., *Molecular Cell*, 2020) (Figure S9). For ribosome, we found a small deviation of ribosomal mRNA number from linear scaling (Figure S8). However, as we show later, our theoretical predictions on the nonlinear scaling of gene expression level still work satisfyingly well in the presence of a small deviation of ribosome from linear scaling.”

We note that our assumption regarding the Michaelis-Menten constant of RNAP is supported by experimental observations that RNAP related genes exhibit constant mRNA concentrations as the cell volume increases (see the following figure, which is shown as Figure S9 in the Supplementary Information).

Figure 1: The RPKM values of RNA polymerase II genes and their average value. The y axis is a good proxy of concentration. There are 52 genes annotated as RNA polymerase II holoenzyme in Gene Ontology database using AmiGO. Error bars represent standard errors. Wilcoxon test results show no significant between-groups differences (V1 vs. V2: $W = 1196$, $p \text{ value} = 3.12e-1$; V2 vs. V3: $W = 1390$, $p \text{ value} = 8.07e-1$; V3 vs. V4: $W = 1357$, $p \text{ value} = 9.77e-1$; V4 vs. V5: $W = 1312$, $p \text{ value} = 7.97e-1$).

For ribosome, we found a small deviation of ribosomal mRNA number from linear scaling using experimental data from Chen et al., *Molecular Cell*, 2020 (Figure S8 in the Supplementary Information). But as the authors mentioned in their paper, the superlinear scaling of ribosomal genes is very likely an artifact due to the drug (1NMPP1) that blocks cell-cycle progress: “the positive size-scaling slope of the ribosomal genes may be a surprising but specific response to 1NMPP1, since it occurs even in CDC28 strains without elutriation.” They found that the superlinear scaling of ribosomal genes is still present in the control experiments while the nonlinearities of other known nonlinear scaling genes are gone.

In the revised version, we have elaborated on this point in the first paragraph of the section “Analysis of experimental data and searching for motifs in the promoter sequences”:

“Interestingly, we found that the ribosomal genes and other translation-related genes, which correspond to the coarse-grained ribosomal genes in our model, are enriched in the superlinear regime with the average β over ribosomal genes about -0.2 . Similar observations were reported in Ref. (Chen et al., *Molecular Cell*, 2020). However, the superlinear scaling of ribosomal genes was also observed in the control cases in which the nonlinearities of other known nonlinear scaling genes were suppressed. Therefore, it was argued that the superlinear scaling of ribosomal genes may be an artifact due to the drug that blocks cell-cycle progress. For RNAP related genes, we found that they

indeed show linear scaling with cell volume as we assume in our coarse-grained model, which is crucial for the linear scaling between the mRNA copy numbers and cell volume (Figure S9). To confirm the validity of our conclusions in the presence of weakly superlinear scaling of ribosomes, we numerically simulated our gene expression model with the recruitment ability of ribosomal gene weaker than the average value and found that even with the small deviation of ribosome number from linear scaling, our theoretical predictions still agree well with the numerical simulations (Figure S10).”

Finally, we remark that even we take account of the small deviation of ribosome number from linear scaling into our simulations, our theoretical predictions regarding the nonlinear degrees of mRNAs and proteins still work satisfyingly well (see the following figure, which is Figure S10 in the Supplementary Information).

Figure 2: Numerical simulations of the full model in which $K_{n,r}$ is larger than $\langle K_{n,i} \rangle$. The nonlinear degree β of the ribosome gene is about -0.2 . (Left) We compare the theoretically predicted nonlinear degrees of mRNA numbers and the measured values from numerical simulations. (Right) We compare the theoretically predicted nonlinear degrees of protein numbers and the measured values from numerical simulations.

4. The most problematic assumption appears to be A3 (“Almost all genes have (approximately) the same ability to bind RNAP.”). As far as I understand, transcriptional regulation – both on a qualitative and on a quantitative level – occurs to a large extent through binding probabilities of RNAP (e.g., Bintu et al. 2005, DOI: 10.1016/j.gde.2005.02.007). Assuming that all genes have pretty much the same binding probabilities thus seems to fly in the face of what is generally assumed in the field. The authors should devote at least a paragraph to discussing this issue, including a survey of relevant experimental data.

Answer: We thank Reviewer 2 for this important comment. In the section “A more realistic model in which genes can have different recruitment abilities”, we would like

to clarify that we consider a continuous distribution of $K_{n,i}$, which does not have to be a narrow distribution. In the simulations of the main text, we chose a lognormal distribution of $K_{n,i}$ with a coefficient of variation (standard deviation/mean) equal to 0.5. In the revised version, we have now included new simulations with a larger coefficient of variation equal to 1 (see the distribution of MM constants in the left panel of the figure below). The new simulations exhibit a broader distribution of nonlinear degrees (see the middle panel of the figure below) that looks more similar to the experimental distributions (Figure 5a in the main text, also shown as Figure 5 in this reply letter in our answer to Comment 5.).

Our theoretical predictions regarding the relationship between the nonlinear degrees and the MM constants still work quite well (see the right panel of the figure below). The middle panel and right panel of the following figures are now included in the manuscript as Figure 4c, d in the main text.

Figure 3: (Left) The lognormal distribution of the Michaelis-Menten constants with a coefficient of variation of 1. (Middle) The resulting distribution of the nonlinear degrees of mRNA. (Right) We compare the theoretically predicted nonlinear degrees of mRNA numbers and the measured one from numerical simulations.

As suggested by Reviewer 2, we have now added a new paragraph in the section “Numerical simulations” to discuss this issue and cited relevant experimental data, including the reference mentioned by Reviewer 2:

“In Figure 4a, b, we set the coefficient of variation (standard deviation/mean) of the MM constants as 0.5. To confirm the validity of model since the recruitment abilities of different promoters can be widely different (Bintu et al., *Current Opinion in Genetics and Development*, 2005, Ong and Corces, *Nature Review Genetics*, 2011, Brewster et al., *Plos Computational Biology*, 2012, Allen and Taatjes, *Nature Review Molecular Cell Biology*, 2015), we also simulate a larger coefficient of variation equal to 1 (Figure 4c,d). The resulting nonlinear degrees exhibit a broader distribution and appear more similar to experiments as we show in the next section. Furthermore, the theoretical predictions of the nonlinear degrees of mRNA numbers still match reasonably well with the simulations.”

5. Fig. 5A shows that both mode and median of the experimental distribution of the “non-linear degree” of cell volume scaling is clearly below 0, i.e., the typical mRNA shows a super-linear scaling with volume. How does that agree with the authors’ model?

Answer: We thank Reviewer 2 for this important comment. We agree that the median value of the nonlinear degrees is slightly negative in the experimental distribution. We would like to clarify that this is consistent with our simulations as we explain the following.

Because sublinear scaling proteins contribute more to the weighted average of the Michaelis-Menten constant, $\langle K_{n,i} \rangle < \overline{K_{n,i}}$, where $\overline{K_{n,i}}$ is the average over all genes with equal weights. However, to have an estimation of the nonlinear degree of most genes, the appropriate MM constant to compare with $\langle K_{n,i} \rangle$ is the median value. For lognormal distribution, we find that the median value of $K_{n,i}$ is close and slightly larger than $\langle K_{n,i} \rangle$. Therefore, the nonlinear degree for the median $K_{n,i}$ is slightly negative compared with the entire distribution of nonlinear degrees, which is what we observed in numerical simulations. Below we show two distributions of the nonlinear degrees of mRNA number from our simulations. In the left panel, the coefficient of variation of $K_{n,i}$ is equal to 0.5, and in the right panel, the coefficient of variation is 1.

Figure 4: (Left) Distribution of the measured nonlinear degrees β of mRNA numbers from numerical simulations. The dashed line marks the location of the median value of the nonlinear degrees. Here, the coefficient of variation of $K_{n,i}$ is equal to 0.5. (Right) The same analysis as the left panel with the coefficient of variation equal to 1.

This is consistent with the experimentally measured nonlinear degrees (see the following figure which is Figure 5a in the main text).

Figure 5: Distribution of the nonlinear scaling degrees of mRNAs of *S. cerevisiae* among genes. The dashed line marks the location of the median value of the nonlinear degrees.

In the revised version of manuscript, we have rephrased the last paragraph in the section “Numerical simulations”:

“Our results suggest that those genes with sublinear scaling and smaller $K_{n,i}$ contribute more in the weighted average of $K_{n,i}$, therefore $\langle K_{n,i} \rangle < \overline{K_{n,i}}$ where $\overline{K_{n,i}}$ is the average over all genes with equal weights. Therefore, genes with $K_{n,i} \approx \overline{K_{n,i}}$ are expected to exhibit superlinear scaling. However, to have an estimation of the nonlinear degree of most genes, the appropriate MM constant to compare with $\langle K_{n,i} \rangle$ is the median value. For the lognormal distribution we used in simulations, we found that the median value of $K_{n,i}$ is close and slightly larger than $\langle K_{n,i} \rangle$. Therefore, the nonlinear degree of the median $K_{n,i}$ is slightly negative compared with the entire distribution (Figure 4a, c), which is consistent with the experimental observations (Figure 5a).”

6. The authors state that “our model predicts that genes with higher (lower) mRNA production rates [than the typical gene] are more likely to exhibit sublinear (superlinear) scaling with cell volumes“. The red points in Fig. 5C show that as a function of the “non-linear degree” (x-axis), the estimated mRNA production rates (y-axis) are minimal at the point where the distribution along the x-axis is most dense (the mode of Fig. 5A) - where the typical gene is located. This appears to contradict the authors interpretation. The authors report an overall correlation between x- and y-axis, which accounts for only $R^2=3\%$ of the observed variation, but do not directly

test their prediction. This omission should be rectified and the result should be discussed appropriately.

Answer: We thank Reviewer 2 for this comment. We would like to clarify that each red point in Figure 5c of the manuscript represents the median values over equal numbers of genes. So even within the dense region of data points, there should be an overall positive correlation. In the revised manuscript, we have also included the Spearman correlation coefficient for the same data, which is 0.35 (see the caption of Figure 5c).

Furthermore, we have now included new simulations taking account of heterogeneous initiation rates to elaborate on the weak but positive correlation between the mRNA production rates and the nonlinear degrees, as also suggested by Reviewer 1.

We note that the recruitment ability not only determines the nonlinear degree of volume scaling but also affects the mRNA production rate since a higher recruitment ability enhances the binding probability of RNAP to the promoter. This suggests that there should be a positive correlation between the mRNA production rate and the nonlinear degree.

Meanwhile, we note that the recruitment ability also depends on the initiation rate $\Gamma_{n,i}$ as $K_{n,i} = \frac{k_{off,i} + \Gamma_{n,i}}{k_{on}}$ (see Eq. (S2) in the Supplementary Information section A). Here i labels the index of gene. A higher initiation rate reduces the recruitment ability (increases the MM constant). For simplicity, in most of our simulations, we consider a constant initiation rate for each gene (except for ribosome and RNAP). In this case, we found a strong positive correlation between the mRNA production rates and the nonlinear degree β (see the left panel of the following figure). However, in a more general model with heterogeneity in the initiation rates, a higher initiation rate increases the gene expression level but also reduces the recruitment ability, and therefore decreases the nonlinear degree. Therefore, heterogeneity in the initiation rates reduces the correlation between the mRNA production rates and the nonlinear degrees.

To confirm this prediction, in the revised version of manuscript, we have now added new simulations using the more general expression of the Michaelis-Menten constant $K_{n,i} = \frac{k_{off,i} + \Gamma_{n,i}}{k_{on}}$. This expression generates a positive correlation between $K_{n,i}$ and $\Gamma_{n,i}$. Details of numerical simulations are now included in a new section of Supplementary Information (section E).

We found that in this case, a heterogeneity in $\Gamma_{n,i}$ indeed reduces the correlation between the gene expression level (quantified by the mRNA production rate) and the nonlinear degree (see the middle panel in the following figure). This also provides a plausible mechanism why the correlation coefficient between the mRNA production rates and nonlinear degrees is weaker in the experimental data (Chen et al., *Molecular Cell*, 2020) compared with our simulations based on the simplified model assuming constant $\Gamma_{n,i}$. We remark that our main conclusion that the nonlinear degree of gene

expression scaling depends on the MM constant $K_{n,i}$ is independent of the correlation between $K_{n,i}$ and $\Gamma_{n,i}$ (the right panel in the following figure).

Figure 6: (Left) We simulate the more general model in which $K_{n,i} = \frac{k_{off,i} + \Gamma_{n,i}}{k_{on}}$. In this panel, $\Gamma_{n,i}$ is constant (except for ribosome and RNAP). The Pearson correlation coefficient between the mRNA production rate (y axis) and the nonlinear degree is 0.86. (Middle) In this panel, we add heterogeneity to $\Gamma_{n,i}$ so that its coefficient of variation (standard deviation/mean) is 1. The Pearson correlation coefficient is reduced to 0.15, close to the experimental value, 0.17 (Figure 5c in the main text). (Right) We confirm that our main results on the relation between the nonlinear degree and the Michaelis-Menten constant is still valid in the presence of heterogeneity in $\Gamma_{n,i}$.

In the new version of manuscript, we have clarified this important point in the second to last paragraph of the section “Numerical simulations”:

“We note that the recruitment ability not only determines the nonlinear degree of volume scaling but also affects the mRNA production rate since a higher recruitment ability enhances the binding probability of RNAP to the promoter. This suggests that there should be a positive correlation between the mRNA production rate and the nonlinear degree. Meanwhile, we note that the recruitment ability also depends on the transcription initiation rate $\Gamma_{n,i}$ (SI A, Eq. (S2)): a higher initiation rate reduces the recruitment ability (increases the MM constant). For simplicity, in most of our simulations, we consider a constant $\Gamma_{n,i}$ for genes (except for ribosome and RNAP), and in this case, we indeed found a strong positive correlation between the mRNA production rates and the nonlinear degree β (Figure S4c). However, in a more general model with heterogeneity in $\Gamma_{n,i}$, a higher initiation rate increases the mRNA production rate but also reduces the recruitment ability so that decreases the nonlinear degree. Therefore, heterogeneity in the initiation rates reduces the correlation between the mRNA production rates and the nonlinear degrees. To confirm this prediction, we also simulate the case of heterogeneous initiation rates (see numerical details in SI E), and our predictions are confirmed numerically (Figure S6). We note that this may be a plausible mechanism of the weak but positive correlation observed in the experimental data, as we discuss in the next section.”

Along with the new section E in Supplementary Information, we have also included the above figures in the Supplementary Information as Figure S6. We appreciate Reviewer 2 for this comment which we believe has significantly improved our manuscript.

7. p.7 "... and then used GSEA to identify those motifs that are en-riched in the nonlinear regime with a threshold p value = $5.00e-2$." Did you correct for multiple testing here?

Answer: Yes, we did. The threshold here is for FDR (false discovery rate) q-values. We apologize for the confusion here. In the new version of manuscript, we have clarified this in the first paragraph of the section "Analysis of experimental data and searching for motifs in the promoter sequences".

8. The relationship of the authors' model to the very similar model developed in the Methods section of Ref. [13] (Eq. (28) onwards) should be discussed in detail.

Answer: We thank Reviewer 2 for this very important comment, and we apologize that we did not give a clear discussion on the relation between our model and the models in the Methods of Ref. [13]. We would like to clarify that there are key differences between these two models:

1. This manuscript focuses on the effects of heterogeneous Michaelis-Menten constants and the resulting nonlinear scaling of gene expression levels, including both mRNA and protein. In contrast, the model in the Methods of Ref. [13] only considers transcription process and homogeneous Michaelis-Menten constants.
2. The main purpose of the model in Ref. [13] is to discuss the effects of nonspecifically bound RNAPs, which is believed to be important in bacterial gene expression (Klumpp and Hwa, *PNAS*, 2008). While the model in our current manuscript mainly consider eukaryotic cells, and the experimental data we analyze is from *Saccharomyces cerevisiae* (Chen et al., *Molecular Cell*, 2020).

In the revised manuscript, we have added new discussions on the relation between this model and the previous one in Ref. [13] in the third paragraph of the Discussion section:

"Our model shares some similarities with the model introduced in the Methods section of Ref. (Lin and Amir, *Nature Communications*, 2018), but also with key differences. This work focuses on the effects of heterogeneous MM constants and the resulting nonlinear scaling of gene expression levels, including both mRNA and protein. In

contrast, the model in the Methods of Ref. (Lin and Amir, *Nature Communications*, 2018) only consider transcription process and homogeneous MM constants. Furthermore, the previous model mainly considers the effects of nonspecifically bound RNAPs, which is believed to be important in bacterial gene expression (Klumpp and Hwa, *PNAS*, 2008). While the model in this work mainly consider eukaryotic cells, and the experimental data we analyze is from *S cerevisiae* (Chen et al., *Molecular Cell*, 2020).”

MINOR COMMENTS

9. p.4 “...if the corresponding proteins have short lifetimes...” The typical lifetimes in yeast appear to be 1/2 of the cell cycle (Belle et al., *PNAS* 2006). Is that short enough? Please discuss.

Answer: We thank Reviewer 2 for this comment, and we have now included new simulations and discussions related to proteins with lifetimes about half of the cell cycle. We note that our theoretical predictions regarding the scaling behaviors of degradable proteins are based on the assumption that both the mRNA and protein lifetimes approach zero. Therefore, deviations are expected for proteins with finite lifetimes. We would like to emphasize that the nature of nonlinear scaling, whether superlinear or sublinear, is nevertheless independent of the protein’s lifetime, as we show in the Methods of the manuscript (Derivation of nonlinear scaling).

We have now added a new figure in the Supplementary Information (Figure S3) to include simulations of proteins with lifetimes about half of the cell cycle (40 mins), which we also show below.

Figure 7: Numerical simulations of the simple model in which all genes share the same recruitment ability to RNAPs except two genes. We show the scaling behaviors of degradable proteins here with lifetime $\tau_{p,i} = 10$ min (a) and $\tau_{p,i} = 40$ min (b). The dashed lines are the theoretical predictions assuming the lifetimes of mRNAs and proteins approach zero.

In the revised manuscript, we have also added discussions related to finite protein lifetimes in the second paragraph of the section “Numerical simulations” and cited the relevant reference (Belle et al., *PNAS*, 2006):

“We note that the lifetimes of degradable proteins can be comparable to the duration of cell cycle, e.g., half of the cell-cycle duration (Belle et al., *PNAS*, 2006), in which case deviations of numerical simulations from our theoretical prediction are expected (Figure S3). The nature of nonlinear scaling, whether superlinear or sublinear, is nevertheless independent of the protein’s lifetime, as we show in Methods.”

10. p.6 “We note that our results do not contradict with Ref. [21]“. How not? Please explain.

Answer: We thank Reviewer 2 for this comment, and we have rephrased our discussion to be more clear and precise. We found that inhibitors were enriched in the sublinear regime, consisting with the results of Ref. [21] (Chen et al., *Molecular Cell*, 2020), but activators were not enriched in the superlinear regime. We note that in Ref. [21], the authors investigated the scaling behaviors of pre-selected activators and inhibitors while we checked all annotated regulators using GSEA, which may be the reason why we obtained a slightly different conclusion. But we would like to remark that the conclusion of Ref. [21] that the interplay of inhibitors and activators can trigger cell-cycle progress is still valid as long as they have different scaling behaviors.

In the revised manuscript, we have rephrased the discussions related to this in the second paragraph of the section “Analysis of experimental data and searching for motifs in the promoter sequences”:

“In Ref. (Chen et al., *Molecular Cell*, 2020), the expression of pre-selected activators for cell cycle were shown to be superlinear, while pre-selected inhibitors were shown to be sublinear. So we next checked the nonlinear degrees of all cell-cycle regulators annotated in the Gene Ontology database (Ashburner et al., *Nature Genetics*, 2000, Carbon et al., *Bioinformatics*, 2008, Carbon et al., *Nucleic Acids Research*, 2020) using GSEA. We found that inhibitors are indeed enriched in the sublinear regime (Figure 5b), but activators were not enriched in the superlinear regime. We remark that the inconsistency may be due to the pre-selection of regulators, but the conclusion of Ref. (Chen et al., *Molecular Cell*, 2020) that the interplay of inhibitors and activators can trigger cell-cycle progress is still valid as long as they have different scaling behaviors.”

11. Methods: I would suggest to restate the definitions of all variables in the Methods section; there are too many of them for the reader to easily memorize these from the main text.

Answer: We thank Reviewer 2 for this comment, and we have added a list of all variables with their definitions in the Methods. Note that we also included a list of key variables with their definitions in Figure 1 of the main text.

Reviewer #3 (Remarks to the Author):

The paper discusses the scaling relation between protein or mRNA concentrations and cell volume during cell growth. While for many genes an approximately linear scaling is seen, some genes deviate from this. The paper proposes a theoretical explanation for both observations, based on the idea that scaling reflects the ability of the genes to recruit RNA polymerases. The latter is similar for most genes, resulting in the linear scaling, but significant deviations can lead to sub- or super-linear scaling.

I quite like the systematic approach that the authors take, starting from a minimal model, extending it to two subpopulations of genes and then to a continuous distribution of promoter affinities. Finally, a comparison with published data for yeast is done.

We thank Reviewer 3 for his/her careful reading and for appreciating our systematic approach. In the revised manuscript, we have made significant changes and included new simulations of periodic cell cycle as suggested by Reviewer 3.

Nevertheless I find the overall approach somewhat unsatisfying as it addresses only a part of the cell cycle. In a balanced growth situation, every protein should double over a cell cycle to make sure the two daughter cells have the same content as the mother cell at the beginning of its cell cycle. Likewise for the volume. In terms of concentration, this means that all concentrations at the end of the cell cycle must be the same as at the beginning. Thus any nonlinear scaling of protein/volume or mRNA/volume can only be transient and a superlinear scaling must be followed by a sub-linear one. These aspects have been discussed for the bacterial case (see ref. 13 and also Bierbaum et al. Phys. Biol. 2015) Here only a part of the process is addressed. I am not against doing so, but it is important to keep in mind that this analysis can be design only give a partial picture.

Answer: We thank Reviewer 3 for this very important comment. In the revised version, we have now included new simulations of periodic cell cycle in which cells grow and divide, which we believe have significantly improved our paper. We choose two degradable proteins, one superlinear and one sublinear, and use the ratio of their concentrations to determine the timing of cell division. When the ratio of their concentrations exceeds some threshold value, the cell divides, in concert with the idea that the ratio of cell-cycle regulators determines cell division (Chen et al., *Molecular Cell*, 2020).

We consider symmetric division so that all mRNAs and proteins are symmetrically distributed between the two daughter cells. For simplicity, we assume constant Michaelis-Menten (MM) constants of RNAP and ribosome binding over time and constant gene copy numbers. Our qualitative results are independent of these assumptions. We track a single lineage of cells so that we monitor one of the daughter cells after cell division. In the simulations, we take the lifetimes of mRNAs and degradable proteins as 5 mins. The sublinear and superlinear proteins that we use as signaling proteins respectively have $K_{n,i} \approx 970$ and $K_{n,i} \approx 2.8 \times 10^4$. The cell divides when the ratio of the superlinear protein concentration and the sublinear

protein concentration exceeds 0.3. We also have an additional requirement on the minimum cell-cycle duration to avoid cell division immediately after cell birth. Other simulation details are the same as Figure 4a, b in the main text and explained in the Supplementary Information section F.

We found that for superlinear genes, their mRNA and protein concentrations decrease initially at the beginning of the cell cycle due to the halved RNAP number at cell birth, but quickly increases as the RNAP number increases (vice versa for sublinear genes). As the cell gets the periodic steady state, all mRNAs and proteins double their numbers at cell division (see the figures below). Regulation of time-dependent MM constants is not required to achieve a robust periodic pattern of gene expression dynamics.

Figure 1: (a) The time trajectory of total protein mass in the periodic cell-cycle simulation. (b) The time trajectories of the two signaling proteins, one superlinear (blue) and one sublinear (red). The cell divides when the ratio of their concentrations exceeds some threshold value. Note that the y axis of the blue curve is multiplied by six to better illustrate the data. (c) The volume-dependence of superlinear (blue) and sublinear (red) mRNA numbers. (d) The same analysis as (c) for non-degradable proteins.

In the revised manuscript, we have added a new section to the Supplementary Information (section F) to discuss the cell-cycle simulations, and the above figures are now included as Figure S11.

We have also added a new paragraph in the Discussion section to discuss the cell cycle simulations and cited the relevant reference (Bierbaum and Klumpp. *Phys. Biol.*, 2015):

“Nonlinear scaling of protein levels is crucial to cell-cycle regulation (Chen et al., *Molecular Cell*, 2020). Time-dependent protein concentrations allow cells to determine the timing of various cell-cycle events, e.g., based on the ratio of two proteins with different scaling behaviors. To confirm this scenario, we have also simulated the case of periodic cell cycle and set the timing of cell division as the protein concentrations of one superlinear protein and one sublinear protein exceeds some threshold value. We found that periodic patterns of mRNA and protein concentration emerge. For superlinear genes, their mRNA and protein concentrations decrease initially at the beginning of cell cycle due to the halved RNAP number at cell

birth, but quickly increases as the RNAP number increases (vice versa for sublinear genes). As the cell gets the periodic steady state, all mRNAs and proteins double their numbers at cell division compared with cell birth (Bierbaum and Klumpp, *Physical Biology*, 2015, Lin and Amir, *Nature Communications*, 2018) (SI F and Figure S11).”

In addition, I have a few more specific comments:

1) Fig 2 is confusing. While I see what you want to do (graphical solution of eq.5), this needs to be explained better, in particular, which parameters are held fixed, what happened to the prefactor of the Michaelis-Menten term?

Answer: We apologize for this confusion. We would like to clarify that all variables (except the fraction of free RNA polymerases F_n) are fixed in the self-consistent equation: $n_c \frac{c_n F_n}{c_n F_n + K_n} = n - n F_n$ (now Eq. 8 in the revised version). These include the total RNAP number n , RNAP concentration c_n , the Michaelis-Menten (MM) constant K_n , and $n_c = \sum_i g_i (1 + \Lambda_{n,i})$. Here g_i is the gene copy number, and $\Lambda_{n,i}$ is the maximum possible number of transcribing RNAPs on one copy of gene i .

The left side of the self-consistent equation represents the number of RNAPs that are bound to promoters or transcribing. The right side represents the difference between the total number of RNAPs and free RNAPs, which should be equal to the left side.

In the revised version, we have emphasized that the intersection of the blue curve and black curve in Figure 2 allows us to find the F_n that solve the self-consistent equation and clarified this in the caption of Figure 2 in the main text, which we also include it here:

Figure 2: Based on the conservation of the total number of RNAPs, a self-consistent equation, as shown in the figure, is derived for the case of homogeneous recruitment abilities to RNAPs (Eq. (8) in Methods). Here n is the total number of RNAPs, $n_c = \sum_i g_i (1 + \Lambda_{n,i})$, and F_n is the fraction of free RNAPs. c_n is the concentration of total RNAPs in the nucleus. All variables except F_n are given. The blue curve and the black line are respectively the left and right sides of the equation shown in the figure. The intersection of the two curves allows us to find the F_n that solves the self-consistent equation. Assuming $K_n/c_n \ll 1$, the fraction of free RNAPs F_n that solves Eq. (8) must be much smaller than 1 if $n < n_c$.”

2) it seems that must be a condition on how n , the number of RNAPs has to grow linearly with the volume for the model to work, which would be a self-consistency requirement. This should in my opinion depend on the degree of binding to DNA. For example for constant initiation rates, a constant free concentration would be required. If all RNAP were free, this would imply a linear scaling with volume (or volume of the nucleus, I am not sure if the latter is assumed to stay constant. If all RNAP is on DNA, such scaling with volume may not be required.

Answer: We would like to clarify that the linear scaling between the number of RNAPs and cell volume is a prediction of our model given the assumption that all the genes share the same Michaelis-Menten constant of RNAP binding, as we discuss in the section “A simplified model in which all genes share the same recruitment ability”. Within this scenario, the protein mass fractions are always constant and largely determined by the gene copy numbers, as we discuss in the Supplementary Information section B.

In the revised manuscript, we have significantly reduced the mathematical derivations and added much more intuitive explanations of our results. We have emphasized that the linear scaling between the RNAP number and cell volume is one of the predictions of our model in the first paragraph of the section “A simplified model in which all genes share the same recruitment ability”:

“In the following, we consider a simplified scenario and assume that (1) the promoters of all genes have the same recruitment ability to RNA polymerases so that $K_{n,i} = K_n$ for all i ; (2) the total RNAP concentration in the nucleus is much larger than K_n , $c_n \gg K_n$ (Bremer et al., *Biochimie*, 2003). As we explain later in this section, the above assumptions lead to two predictions: (1) almost all RNAPs are bound to a promoter or transcribing; (2) the protein mass fractions are constant over time. Therefore the protein numbers of all genes are proportional to the cell volume, including RNAP. From prediction (1), it follows that the total mRNA production rate of all genes is proportional to the total number of RNAPs. From prediction (2), it follows that this rate is also proportional to the cell volume. Finally, combined with assumption (1), it follows that the total mRNA production rate should be evenly distributed among all genes. Therefore, all genes' mRNA numbers increase linearly with volume, which is the main result of this section.”

In the section “A more realistic model in which genes can have different recruitment abilities”, we agree that the constant total RNAP concentration is an important assumption and we have now added a paragraph to clarify the assumptions we make in the case of continuously distributed MM constants:

“We remark that to ensure the linear scaling of the majority of genes, the RNAP number and the ribosome number should be proportional to the cell volume, which requires that the MM constants of RNAP and ribosome are close to the average value. These are the additional assumptions we make in the case of continuously distributed $K_{n,i}$. For RNAP, it is supported by the constant mRNA concentrations of RNAP related genes observed in the experimental data from Ref. (Chen et al., *Molecular Cell*, 2020) (Figure S9). For ribosome, we found a small deviation of ribosomal mRNA number from linear scaling (Figure S8). However, as we show later, our theoretical predictions on the nonlinear scaling of gene expression level still work satisfyingly well in the presence of a small deviation of ribosome from linear scaling.”

We note that our assumption regarding the Michaelis-Menten constant of RNAP is supported by experimental observations that RNAP related genes exhibit constant mRNA concentrations as the cell volume increases (see the following figure, which is shown as Figure S9 in the Supplementary Information).

Figure 3: The RPKM values of RNA polymerase II genes and their average value. The y axis is a good proxy of concentration. There are 52 genes annotated as RNA polymerase II holoenzyme in Gene Ontology database using AmiGO. Error bars represent standard errors. Wilcoxon test results show no significant between-groups differences (V1 vs. V2: $W = 1196$, $p \text{ value} = 3.12e-1$; V2 vs. V3: $W = 1390$, $p \text{ value} = 8.07e-1$; V3 vs. V4: $W = 1357$, $p \text{ value} = 9.77e-1$; V4 vs. V5: $W = 1312$, $p \text{ value} = 7.97e-1$).

Finally, we would like to clarify that within our model, we assume that the nuclear volume is proportional to the cell volume, which is supported by experimental observations (Newman and Nurse, *The Journal of Cell Biology*, 2007).

3) likewise ribosomes should also be in the normal group with linear scaling, which seems not to be the case in the experimental data. The authors claim that this does not change their simulation results, but I do not see this. They refer to Fig. S8B, which I think needs to be compared to Fig. S4B and these appear pronouncedly different to me.

Answer: We thank Reviewer 3 for this very important comment. First, we would like to clarify that the superlinear scaling of ribosomal genes in the experimental paper is likely an experimental artifact due to the drugs that block cell-cycle progress (1NMPP1). The authors of the original experimental paper (Chen et al., *Molecular Cell*, 2020) mentioned in their paper that: “the positive size-scaling slope of the ribosomal genes may be a surprising but specific response to 1NMPP1, since it occurs even in CDC28 strains without elutriation.” They found that the superlinear scaling of ribosomal genes is still present in the control experiments while the nonlinearities of other known nonlinear scaling genes are gone.

In the revised version, we have elaborated on this point in the first paragraph of the section “Analysis of experimental data and searching for motifs in the promoter sequences”:

“Interestingly, we found that the ribosomal genes and other translation-related genes, which correspond to the coarse-grained ribosomal genes in our model, are enriched in the superlinear regime with the average β over ribosomal genes about -0.2 . Similar observations were reported in Ref. (Chen et al., *Molecular Cell*, 2020). However, the superlinear scaling of ribosomal genes was also observed in the control cases in which the nonlinearities of other known nonlinear scaling genes were suppressed. Therefore, it was argued that the superlinear scaling of ribosomal genes may be an artifact due to the drug that blocks cell-cycle progress. For RNAP related genes, we found that they indeed show linear scaling with cell volume as we assume in our coarse-grained model, which is crucial for the linear scaling between the mRNA copy numbers and cell volume (Figure S9). To confirm the validity of our conclusions in the presence of weakly superlinear scaling of ribosomes, we numerically simulated our gene expression model with the recruitment ability of ribosomal gene weaker than the average value and found that even with the small deviation of ribosome number from linear scaling, our theoretical predictions still agree well with the numerical simulations (Figure S10).”

Finally, we have improved our fitting protocol to find the nonlinear degrees of proteins (see the detailed answer to Comment 7). We found that the measured nonlinear degrees in the case of weakly superlinear ribosomes are indeed similar to the case of linear ribosomes (see Figure 7 in this reply letter in the answer to Comment 7).

4) changes in transcription factor concentrations could provide an alternative explanation for nonlinear scaling. Why can that be completely neglected?

Answer: We thank Reviewer 3 for this comment. Within our model, we think that the changing transcription factor (TF) concentrations can be modelled as a volume-dependent or time-dependent Michaelis-Menten constant. We completely agree with Reviewer 3 that this could be alternative mechanisms to achieve a nonlinear scaling. In this work, we choose to focus on a simple scenario in which genes' Michaelis-Menten constants to RNAP binding are different but do not change over time.

We would like to mention that in our GSEA (Gene Set Enrichment Analysis) analysis, we found that TF related terms were not enriched in the nonlinear regime (see the figure below for the annotated functional gene sets in KEGG that are enriched in the nonlinear scaling regime, which is shown as Figure S8 in the Supplementary Information), which means TFs do not change their concentrations in general.

Figure 4: Functional gene sets enriched in the nonlinear scaling regime. GeneRatio represents tags in GSEA, which is the fraction of leading-edge genes in those genes that are both in our list of genes and in the corresponding functional gene sets of KEGG. Point size represents the number of leading-edge genes. Colors of the points represent the adjusted p value (FDR). Names of the gene sets are followed by their IDs in KEGG data base (Kanehisa and Goto, *Nucleic Acids Research*, 2000, Kanehisa, *Protein Science*, 2019, Kanehisa et al., *Nucleic Acids Research*, 2020).

Therefore, we argue that a changing TF concentration could be more specific to certain genes instead of a general situation. In the revised manuscript, we have clarified this point in the Discussion section:

“We note that other mechanisms of nonlinear scaling of gene expression levels are possible, such as time-dependent transcription factor concentrations (Chen et al., *Molecular Cell*, 2020), or time-dependent initiation rates. A time-dependent transcription factor concentration is equivalent to a time-dependent MM constant $K_{n,i}$ within our model. However, we note that our GSEA analysis showed that TF related terms were not enriched in the nonlinear regime (Figure S8), which means TFs do not change their concentrations in general. Therefore, we argue that a changing TF concentration is more specific to certain genes instead of a general situation. Also, we remark that our model does not require time-dependent variables to achieve changing concentrations, and the changing concentrations of mRNAs and proteins are the result of the competition between genes to the limiting resource of RNAPs.”

5) p. 5 right column, top: why is the initial mass fraction used as the weight? One could also imagine other quantities to be used, e.g. the average mass fraction

Answer: We thank Reviewer 3 for this comment. In the revised manuscript, we have also used the average mass fraction as the weight to compute the average Michaelis-Menten constant. We find that this alternative weight works equally well (see the figures below).

Figure 5: We compare the theoretically predicted nonlinear degrees of mRNA numbers and the measured one from numerical simulations. In both panels, the coefficient of variation of the MM constants is 1. (Left) The average MM constant is computed as a weighted average over the initial protein mass fractions. (Right) The same simulations as the left panel, but the weight is based on the time-averaged protein mass fractions over the total duration of simulations.

The above figure is now included in the Supplementary Information as Figure S5.

In the revised manuscript, we have also added an argument that the substitution of K_n by the weighted average is a good approximation. We have clarified this point in the

third paragraph of the section “A more realistic model in which genes can have different recruitment abilities”:

“In this case, we propose that the nonlinear scaling, Eq. (4), is still approximately valid for any gene if K_n is replaced by $\langle K_{n,i} \rangle$, the average value of $K_{n,i}$ among all genes with some appropriate weights. In SI D, we show that the appropriate weight can be well approximated by the protein mass fractions, as we confirm numerically in the next section.”

The argument is now included in the Supplementary Information section D.

6) fig. 5c: the correlation seen here is very weak. I think it would be good to check with the model whether such weak correlation is sufficient in the model or whether the model leads to stronger correlations.

Answer: We thank Reviewer 3 for this important comment and similar comments are also raised by Reviewer 1 and 2. In the revised manuscript, we have now included new simulations taking account of heterogeneous initiation rates to elaborate on the weak but positive correlation between the mRNA production rates and the nonlinear degrees.

We note that the recruitment ability not only determines the nonlinear degree of volume scaling but also affects the mRNA production rate since a higher recruitment ability enhances the binding probability of RNAP to the promoter. This suggests that there should be a positive correlation between the mRNA production rate and the nonlinear degree.

Meanwhile, we note that the recruitment ability also depends on the initiation rate $\Gamma_{n,i}$ as $K_{n,i} = \frac{k_{off,i} + \Gamma_{n,i}}{k_{on}}$ (see Eq. (S2) in the Supplementary Information section A). Here i labels the index of gene. A higher initiation rate reduces the recruitment ability (increases the MM constant). For simplicity, in most of our simulations, we consider a constant initiation rate for each gene (except for ribosome and RNAP). In this case, we found a strong positive correlation between the mRNA production rates and the nonlinear degree β (see the left panel of the following figure). However, in a more general model with heterogeneity in the initiation rates, a higher initiation rate increases the gene expression level but also reduces the recruitment ability, and therefore decreases the nonlinear degree. Therefore, heterogeneity in the initiation rates reduces the correlation between the mRNA production rates and the nonlinear degrees.

To confirm this prediction, in the revised manuscript, we have now added new simulations using the more general expression of the Michaelis-Menten constant

$K_{n,i} = \frac{k_{off,i} + \Gamma_{n,i}}{k_{on}}$. This expression generates a positive correlation between $K_{n,i}$ and $\Gamma_{n,i}$. Details of numerical simulations are now included in a new section of Supplementary Information (section E).

We found that in this case, a heterogeneity in $\Gamma_{n,i}$ indeed reduces the correlation between the gene expression level (quantified by the mRNA production rate) and the nonlinear degree (see the middle panel in the following figure). This also provides a plausible mechanism why the correlation coefficient between the mRNA production rates and nonlinear degrees is weaker in the experimental data (Chen et al., *Molecular Cell*, 2020) compared with our simulations based on the simplified model assuming constant $\Gamma_{n,i}$. We remark that our main conclusion that the nonlinear degree of gene expression scaling depends on the MM constant $K_{n,i}$ is independent of the correlation between $K_{n,i}$ and $\Gamma_{n,i}$ (the right panel in the following figure).

Figure 6: (Left) We simulate the more general model in which $K_{n,i} = \frac{k_{off,i} + \Gamma_{n,i}}{k_{on}}$. In this panel, $\Gamma_{n,i}$ is constant (except for ribosome and RNAP). The Pearson correlation coefficient between the mRNA production rate (y axis) and the nonlinear degree is 0.86. (Middle) In this panel, we add heterogeneity to $\Gamma_{n,i}$ so that its coefficient of variation (standard deviation/mean) is 1. The Pearson correlation coefficient is reduced to 0.15, close to the experimental value, 0.17 (Figure 5c in the main text). (Right) We confirm that our main results on the relation between the nonlinear degree and the Michaelis-Menten constant is still valid in the presence of heterogeneity in $\Gamma_{n,i}$.

In the new version of manuscript, we have clarified this important point in the second to last paragraph of the section ‘‘Numerical simulations’’:

‘‘We note that the recruitment ability not only determines the nonlinear degree of volume scaling but also affects the mRNA production rate since a higher recruitment ability enhances the binding probability of RNAP to the promoter. This suggests that there should be a positive correlation between the mRNA production rate and the nonlinear degree. Meanwhile, we note that the recruitment ability also depends on the transcription initiation rate $\Gamma_{n,i}$ (SI A, Eq. (S2)): a higher initiation rate reduces the recruitment ability (increases the MM constant). For simplicity, in most of our

simulations, we consider a constant $\Gamma_{n,i}$ for genes (except for ribosome and RNAP), and in this case, we indeed found a strong positive correlation between the mRNA production rates and the nonlinear degree β (Figure S4c). However, in a more general model with heterogeneity in $\Gamma_{n,i}$, a higher initiation rate increases the mRNA production rate but also reduces the recruitment ability so that decreases the nonlinear degree. Therefore, heterogeneity in the initiation rates reduces the correlation between the mRNA production rates and the nonlinear degrees. To confirm this prediction, we also simulate the case of heterogeneous initiation rates (see numerical details in SI E), and our predictions are confirmed numerically (Figure S6). We note that this may be a plausible mechanism of the weak but positive correlation observed in the experimental data, as we discuss in the next section.”

Along with the new section E in Supplementary Information, we have also included the above figures in the Supplementary Information as Figure S6. We appreciate Reviewer 3 for pointing this, which we believe has significantly improved our manuscript.

7) fig. S4B: there seems to be a systematic deviation between the simulation and the prediction, for proteins much more than for the mRNA (Fig. 4D). Why is that?

Answer: We apologize for this issue, and we think that the deviation is largely due to the problem in our previous fitting protocol on the volume dependence of protein number. In the revised manuscript, we have now improved our fitting protocol and obtained a better agreement between the theoretical predictions and numerical simulations.

In the previous version of our manuscript, we first Taylor-expanded the formula of protein number $\Delta p_i = C_i \ln(1 + \alpha_i \Delta V)$ for small cell volume change ΔV . We then fit the simulation data to the expanded formula, and obtained the nonlinear degree α_i . We recently improved our fitting protocol by directly fitting the simulation data to the original formula using the built-in fitting function of MATLAB.

After implementing the new protocol, we found that the measured nonlinear degrees of proteins agree well with the theoretical predictions (see the left panel of the following figure). Furthermore, for the case of weakly superlinear ribosomal genes (see the answer to Comment 3), the agreement is also quite well (see the right panel of the following figure).

Figure 7: (Left) We compare the theoretically predicted nonlinear degrees of non-degradable protein and the measured values from numerical simulations. In this panel, the ribosomal gene has a linear scaling with cell volume. (Right) We compare the theoretically predicted nonlinear degrees of non-degradable protein and the measured values from numerical simulations. In this panel, the ribosomal gene has a superlinear scaling with cell volume. The nonlinear degree β of the ribosome gene is about -0.2 .

In the revised manuscript, we have updated the figures (Figure S4d and Figure S10b in the Supplementary Information) related to the fitting of non-degradable protein.

REVIEWER COMMENTS

Reviewer #1 (Remarks to the Author):

In the revised version of their manuscript, Lin and Wang rewrote large parts of the manuscript according to the suggestions of all 3 reviewers. In addition to including specific additional analyses requested, the new version is now much more upfront about assumptions made in the different sections, and further improves the accessibility of the manuscript by adding intuitive explanations of the results. The authors addressed all comments by the reviewers – either by adding new results, or at least by being clearer about the limitations of their assumptions. Overall, I find the revised version much improved.

As stated in my assessment of the initial manuscript, I appreciate the effort of the authors to provide a mathematical framework to understand the scaling of transcript and protein concentrations with cell size. While conceptually the idea that limiting polymerase accounts for the increase of transcription rate with cell size, and that different 'recruitment abilities' of polymerase can then lead to different size scaling for different genes, has been brought forward before (e.g. work by the Marguerat, Raj and Skotheim labs, as well as Heldt et al., which already includes a mathematical model), Lin and Wang provide a more general framework. Even in the 'general' scenario discussed in this manuscript, strong (but reasonable, and now also well-communicated) assumptions are made. However, the framework itself will also allow future studies to explore alternative situations, which makes the manuscript a relevant contribution to the field.

Reviewer #2 (Remarks to the Author):

GENERAL REMARKS

The authors have adequately responded to most of my previous comments. However, I remain unconvinced that the strong assumptions are warranted, and I still find the agreement with experimental data questionable, in particular the weak correlation between nonlinear degree and mRNA production rate (Fig. 5c). I think the authors provide a nice theoretical model. However, I find the agreement with data too weak to conclude that the model is an adequate description of reality.

SPECIFIC COMMENTS

1. I do not find the "intuitive" explanation of the authors in "A simplified model in which all genes share the same recruitment ability" very intuitive, even if it is partly similar to the one I supplied with my previous review. For an intuitive understanding, the self-consistency equation illustrated in Fig. 2 is not necessary – the authors assume, based on $c_n/K_n \sim 10$ [33], that the fraction of free RNAP is negligible, and so they set the mRNA production rate proportional to the total RNAP concentration.
2. Line 151: "the above assumptions lead to two predictions: (1) almost all RNAPs are bound to a promoter or transcribing; (2) the protein mass fractions are constant over time. Therefore the protein numbers of all genes are proportional to the cell volume, including RNAP." – The last sentence does not follow from the previous sentences. For this inference, one also needs to assume that the total protein concentration is proportional to volume (which is reasonable, but does not follow from the authors' model).
3. Line 162: "Therefore, all genes' mRNA numbers increase linearly with volume, which is the main result of this section." – This "main result" is a trivial consequence of assuming that (i) there is no free RNAP, (ii) all promoters bind RNAP with the same probability, and (iii) total protein concentration is constant across the cell cycle. While all of this is reasonable, it could be explained more clearly and

consistently.

4. Line 165: "We now argue the validity of prediction (1). ..." Really, prediction (1) follows directly from $c_n/K_n \gg 1$ – no self-consistency equation is needed (see my point 1).

5. In their response to my previous comment 2, the authors state: "We would like to clarify that the linear scaling between the RNAP, ribosome number, and cell volume is a prediction of our model given the assumption that all the genes share the same Michaelis-Menten constant of RNAP binding". This is wrong. The authors' model assumes that RNAP and ribosome both scale like the "average" protein, and thus a linear scaling between RNAP and ribosome number is built into the model. There is nothing that links that to volume – unless one assumes that total protein concentration is constant across the cell cycle (see my comment 2). Which the authors apparently do implicitly, but need to do explicitly.

6. My strongest remaining concern relates to my previous comment 6, about the disagreement of the correlation between non-linear scaling and mRNA production rate (Fig. 5c). This correlation is the strongest prediction of the authors' model, and it is rather weak in the observed data (Spearman's $\rho^2 = 12\%$). The authors argue that additional complications might have reduced the correlation in the experimental data, but then how do we know if the model is appropriate or not? Why do the authors only assess linear correlations, but do not compare the predictions (Fig. 4) directly to the data?

Reviewer #3 (Remarks to the Author):

The authors have done a lot of additional work which clarifies many of the issues raised by the reviewers. I am not entirely convinced of their argument that transcription factors can be excluded as a source of the nonlinear scaling (as proposed by Chen et al. in ref. 21), but at least there is an argument.

One issue that is not convincing however, is the argument that $c_n \gg K_n$. First of all, ref. 33 (Bremer et al.), to which the authors refer in their manuscript and multiple times in their reply to the reviewers deals with the free cytoplasmic RNAP concentration in E. coli. I doubt that one can use this to make arguments on the RNAP concentration in the nucleus of yeast. Moreover, even for E. coli, this point was debated (cf. refs 33 and 23).

In my opinion, additional revision is needed.

List of main changes:

1. As suggested by Reviewer 2, we have rewritten the first half of the section “A simplified model in which all genes share the same recruitment ability” to be more explicit about our model assumptions.
2. We have included new simulations of a modified model to support the validity of our model.
3. We have clarified that the main conclusions of our work are independent of the assumption that the total RNAP polymerase concentration is much larger than the typical Michaelis-Menten constant.

Reviewer #1 (Remarks to the Author):

In the revised version of their manuscript, Lin and Wang rewrote large parts of the manuscript according to the suggestions of all 3 reviewers. In addition to including specific additional analyses requested, the new version is now much more upfront about assumptions made in the different sections, and further improves the accessibility of the manuscript by adding intuitive explanations of the results. The authors addressed all comments by the reviewers – either by adding new results, or at least by being clearer about the limitations of their assumptions. Overall, I find the revised version much improved.

As stated in my assessment of the initial manuscript, I appreciate the effort of the authors to provide a mathematical framework to understand the scaling of transcript and protein concentrations with cell size. While conceptually the idea that limiting polymerase accounts for the increase of transcription rate with cell size, and that different ‘recruitment abilities’ of polymerase can then lead to different size scaling for different genes, has been brought forward before (e.g. work by the Marguerat, Raj and Skotheim labs, as well as Heldt et al., which already includes a mathematical model), Lin and Wang provide a more general framework. Even in the 'general' scenario discussed in this manuscript, strong (but reasonable, and now also well-communicated) assumptions are made. However, the framework itself will also allow future studies to explore alternative situations, which makes the manuscript a relevant contribution to the field.

We thank Reviewer 2 for their careful reading and appreciation of the general framework we provide. We completely agree with Reviewer 1 that the framework offers a platform to explore alternative situations, making our work important and useful to the field.

Reviewer #2 (Remarks to the Author):

GENERAL REMARKS

The authors have adequately responded to most of my previous comments. However, I remain unconvinced that the strong assumptions are warranted, and I still find the agreement with experimental data questionable, in particular the weak correlation between nonlinear degree and mRNA production rate (Fig. 5c). I think the authors provide a nice theoretical model. However, I find the agreement with data too weak to conclude that the model is an adequate description of reality.

SPECIFIC COMMENTS

1. I do not find the “intuitive” explanation of the authors in “A simplified model in which all genes share the same recruitment ability” very intuitive, even if it is partly similar to the one I supplied with my previous review. For an intuitive understanding, the self-consistency equation illustrated in Fig. 2 is not necessary – the authors assume, based on $c_n/K_n \sim 10$ [33], that the fraction of free RNAP is negligible, and so they set the mRNA production rate proportional to the total RNAP concentration.

Answer: We agree with Reviewer 2 that the self-consistent equation illustrated in Fig. 2 can be omitted. We have removed Fig. 2 from the main text in the revised manuscript and put it to the Supplementary Information. We have also rewritten the first half of the same section to be more explicit about the assumption that the fraction of free RNAPs is negligible.

2. Line 151: “the above assumptions lead to two predictions: (1) almost all RNAPs are bound to a promoter or transcribing; (2) the protein mass fractions are constant over time. Therefore the protein numbers of all genes are proportional to the cell volume, including RNAP.” – The last sentence does not follow from the previous sentences. For this inference, one also needs to assume that the total protein concentration is proportional to volume (which is reasonable, but does not follow from the authors’ model).

Answer: We completely agree with Reviewer 2 that an additional assumption that the total protein number is proportional to cell volume is needed. In the revised manuscript, we have emphasized this point in the last paragraph of the section “Model of gene expression at the whole-cell level” and cited relevant references. We also mention it again in the first paragraph of the section “A simplified model in which all genes share the same recruitment ability.”

3. Line 162: “Therefore, all genes’ mRNA numbers increase linearly with volume, which is the main result of this section.” – This “main result” is a trivial consequence

of assuming that (i) there is no free RNAP, (ii) all promoters bind RNAP with the same probability, and (iii) total protein concentration is constant across the cell cycle. While all of this is reasonable, it could be explained more clearly and consistently.

Answer: We thank Reviewer 2 for this helpful comment and agree that we can make the explanation more clear and consistent. In the revised manuscript, we have rewritten the section “A simplified model in which all genes share the same recruitment ability” as suggested by Reviewer 2. We now emphasize that our main conclusion that all genes’ mRNA numbers increase linearly with volume follows directly from the three assumptions Reviewer 2 mentioned.

4. Line 165: “We now argue the validity of prediction (1). ...” Really, prediction (1) follows directly from $c_n/K_n \gg 1$ – no self-consistency equation is needed (see my point 1).

Answer: We completely agree with Reviewer 2 that it is intuitive that if the concentration of total RNAPs is much larger than the Michaelis-Menten constant, most RNAPs will be binding to promoters or transcribing. In the revised manuscript, we have made it clear that the assumption that most RNAPs are bound to a promoter or transcribing follows from $c_n \gg K_n$ and removed the self-consistent equation from the main text.

We also apologize that we didn’t explain well and would like to mention that the binding probability of RNAPs to a promoter depends on the concentration of free RNAPs, $P_b = \frac{c_{n,free}}{c_{n,free} + K_n}$, instead of total RNAPs (c_n). A negligible fraction of free RNAPs also requires that the total RNAP number (n) is smaller than the threshold value n_c . If $n > n_c$, the fraction of free RNAPs can be non-negligible, and genes start to be saturated by extra RNAPs (Lin and Amir, *Nature Communications*, 2018), which we also discuss in the Supplementary Information section B.

5. In their response to my previous comment 2, the authors state: “We would like to clarify that the linear scaling between the RNAP, ribosome number, and cell volume is a prediction of our model given the assumption that all the genes share the same Michaelis-Menten constant of RNAP binding”. This is wrong. The authors’ model assumes that RNAP and ribosome both scale like the “average” protein, and thus a linear scaling between RNAP and ribosome number is built into the model. There is nothing that links that to volume – unless one assumes that total protein concentration is constant across the cell cycle (see my comment 2). Which the authors apparently do implicitly, but need to do explicitly.

Answer: We thank Reviewer 2 for this very helpful comment. In the revised manuscript, we have explicitly explained that we assume the total protein

concentration is constant across the cell cycle in the last paragraph of the section “Model of gene expression at the whole-cell level” and cited relevant references.

6. My strongest remaining concern relates to my previous comment 6, about the disagreement of the correlation between non-linear scaling and mRNA production rate (Fig. 5c). This correlation is the strongest prediction of the authors’ model, and it is rather weak in the observed data (Spearman’s $\rho^2 = 12\%$). The authors argue that additional complications might have reduced the correlation in the experimental data, but then how do we know if the model is appropriate or not? Why do the authors only assess linear correlations, but do not compare the predictions (Fig. 4) directly to the data?

Answer: We thank Reviewer 2 for this very helpful comment. In the revised manuscript, we have included a direct comparison between simulations and experimental data. We find that the simulated distribution of nonlinear scaling degrees matches the experimentally measured distribution reasonably well (see the following figure).

Figure 1: We compare the numerically simulated distribution of nonlinear degrees (circles) and the experimentally measured distribution. In the simulation, the distribution of the Michaelis-Menten constants follows a lognormal distribution with a coefficient of variation equal to 0.8. Other simulation details are included in the Supplementary Information Figure S8.

This figure is now included in the Supplementary Information as Figure S8. We would like to mention that since we don’t have data of parameters such as the Michaelis-Menten constant $K_{n,i}$, we are not able to test the relation between the nonlinear degrees and the Michaelis-Menten constants, which can be improved in the future when more data are available.

We also would like to clarify that adding heterogeneous initiation rates to our model reduces the correlation between the nonlinear scaling and mRNA production rates but does not affect our main conclusion that the heterogeneous recruitment abilities determine the nonlinear scaling. Our theoretical prediction on the relation between the nonlinear degrees and the Michaelis-Menten constant ($\beta_i = -\frac{(K_{n,i} - \langle K_{n,i} \rangle) n(0)}{K_{n,i} n_c}$, Eq. 17 in Methods) is equally valid, in which the initiation rate ($\Gamma_{n,i}$) is included in the Michaelis-Menten constant, $K_{n,i} = \frac{k_{off,i} + \Gamma_{n,i}}{k_{on}}$ (see the following figure).

Figure 2: (Left) We simulate the more general model in which $K_{n,i} = \frac{k_{off,i} + \Gamma_{n,i}}{k_{on}}$. In this panel, we add heterogeneity to $\Gamma_{n,i}$ so that its coefficient of variation (standard deviation/mean) is 1. The Pearson correlation coefficient is reduced to 0.18, close to the experimental value, 0.17 (Figure 4c in the main text). (Right) We confirm that our main results on the relation between the nonlinear degree and the Michaelis-Menten constant are still valid in the presence of heterogeneity in $\Gamma_{n,i}$.

The above figures are in the Supplementary Information as Figure S6(b, c).

Finally, we would like to clarify that several pieces of evidence support the validity of our model.

1. Regarding the correlation between the nonlinear scaling and mRNA production, we would like to argue that the correlation coefficient is small but significant enough to support the validity of our model. To show this, we simulate a modified model to mimic a scenario where the nonlinear scaling has nothing to do with the different recruitment abilities. We consider two sets of Michaelis-Menten constants such that they share the same randomness of the heterogeneous initiation rates $K_{n,i} = \frac{k_{off,i} + \Gamma_{n,i}}{k_{on}}$, but their random off-rates $k_{off,i}$ are independent of each other. The simulation is the same as Figure 2 in this reply letter, except that the nonlinear degrees and the mRNA production rates are calculated using the two sets of $K_{n,i}$ respectively to simulate the case that the nonlinear scaling is independent of the Michaelis-Menten constants. We

find that the Pearson correlation coefficient between the nonlinear degrees and mRNA production rates becomes negative in the modified model. However, the original model and the experimental data both exhibit positive correlations. We repeat the simulation multiple times and find that the correlation coefficient of the modified model is always smaller than that of the original model (see the following figure).

Figure 3: (Left) The positive Pearson correlation between the mRNA production rates and the nonlinear scaling for the model in which the recruitment abilities affect the nonlinear scaling. (Middle) The negative Pearson correlation between the mRNA production rates and the nonlinear scaling for the model in which the recruitment abilities do not affect the nonlinear scaling. (Right) We repeat the simulations multiple times and compare the Pearson correlation coefficients of the two models. The correlation coefficients of the model in which the recruitment ability does not influence the nonlinear scaling (ρ_2) are negative and always smaller than those of our model (ρ_1).

The above figure is now included in the Supplementary Information as Figure S6(b, d, e). In the revised manuscript, we have also added related discussions in the second paragraph of the section “Analysis of experimental data and searching for motifs in the promoter sequences.”

2. In the analysis of promoter sequences, we find that the binding motifs of transcription factors that exhibit positive regulation are enriched in the sublinear genes, suggesting that the sublinear genes tend to have higher recruitment abilities to RNAPs, which is also consistent with our conclusion.

Reviewer #3 (Remarks to the Author):

The authors have done a lot of additional work which clarifies many of the issues raised by the reviewers. I am not entirely convinced of their argument that transcription factors can be excluded as a source of the nonlinear scaling (as proposed by Chen et al. in ref. 21), but at least there is an argument.

One issue that is not convincing however, is the argument that $c_n \gg K_n$. First of all, ref. 33 (Bremer et al.), to which the authors refer in their manuscript and multiple times in their reply to the reviewers deals with the free cytoplasmic RNAP concentration in E. coli. I doubt that one can use this to make arguments on the RNAP concentration in the nucleus of yeast. Moreover, even for E. coli, this point was debated (cf. refs 33 and 23).

In my opinion, additional revision is needed.

Answer: We thank Reviewer 3 for their careful reading of our revised manuscript and this very useful comment. We agree that we need to be more explicit about the assumption that $c_n \gg K_n$. We would like to clarify that our main conclusions of this work on the relationship between the nonlinear scaling and the recruitment abilities do not rely on the above assumption. The primary purpose of the assumption that $c_n \gg K_n$ is to make the condition of a negligible fraction of free RNA polymerases (F_n) more mathematically well defined as $\frac{n}{n_c} < 1$. Since we mainly focus on the scenario in which RNAP is limiting and $F_n \ll 1$, the transition details from RNAP limiting phase to gene limiting phase is not important to our conclusions.

In the revised manuscript, we have now explicitly explained this in the section “Details of the gene expression model” in Methods. We also agree with Reviewer 3 that although this assumption $c_n \gg K_n$ is biologically reasonable, it remains to be tested in yeast, which we have also mentioned explicitly in the revised manuscript.

REVIEWERS' COMMENTS

Reviewer #2 (Remarks to the Author):

The authors have adequately responded to my previous comments.

MINOR COMMENT

I would recommend that the authors double check all references to Supplementary Figures; e.g., the references to Supplementary Figures 8 and 9 in lines 266-269 need to be updated.

Reviewer #3 (Remarks to the Author):

In my opinion, the authors have clarified the remaining issues. Making the model assumptions more explicit is definitely helpful. No further revision needed.

Reviewer #2 (Remarks to the Author):

The authors have adequately responded to my previous comments.

MINOR COMMENT

I would recommend that the authors double check all references to Supplementary Figures; e.g., the references to Supplementary Figures 8 and 9 in lines 266-269 need to be updated.

We thank Reviewer 2's careful reading and helpful comments, which we believe have significantly improved our paper. We have now double-checked the references to the Supplementary Figures as suggested.

Reviewer #3 (Remarks to the Author):

In my opinion, the authors have clarified the remaining issues. Making the model assumptions more explicit is definitely helpful. No further revision needed.

We thank Reviewer 3's careful reading and helpful comments, which we believe have significantly improved our paper.